# High-frequency, low-energy organic event-based sensors for closed-loop neurostimulation

Chi-Yuan Yang[1,11], Zifang Zhao [2,3,11], Han-Yan Wu[1,11], Dace Gao[1], Jun-Da Huang[1], Junpeng Ji[1], Miao Xiong[1], Tiefeng Liu[1,4], Padinhare C. Harikesh[1], Adam Marks[5], Xin-Yi Wang[6], Matteo Massetti[1], Shan Shao[7], Jian Pei[6], Iain McCulloch[5,8], Magnus Berggren[1,4], Deyu Tu[1], Jennifer Gelinas[7,9], Dion Khodagholy [10] ✉ & Simone Fabiano [1,4] ✉

Event-based bioelectronic sensors enable real-time detection and modulation of neural activity. However, conventional silicon interfaces are rigid and energy intensive, whereas organic electrochemical neuron (OECN)-based sensors, though promising, have been limited by slow firing rates, high energy use and scalability challenges. Here we present an OECN-based sensor capable of rapid, energy-efficient neural signal detection for closed-loop neurostimulation. These event-driven sensors respond within ~1 ms and generate voltage pulses up to 1.1 kHz, covering the full bandwidth of mammalian neuronal activity (0.5–1,000 Hz) while consuming only ~40 pJ per spike. Accurate detection of hippocampal interictal epileptiform discharges is demonstrated. Integrated with microelectrodes, these OECN-based sensors enable closed-loop neuromodulation by delivering real-time stimulation to suppress pathological sleep spindle oscillations in vivo. Combining biorealistic operation with ultra-low energy use, OECN-based sensors are good candidates for the next generation of implantable bioelectronics in energy-constrained environments.

Bioelectronic devices bridge the gap between biological and artificial systems, enabling the detection, processing and modulation of physiological signals[1]. These technologies have transformed medical diagnostics and therapeutic interventions, offering solutions for neurological disorders, cardiovascular monitoring and brain–computer interfaces[2]. Among these, neural interfaces are critical in studying brain function and developing treatments for conditions such as epilepsy, Parkinson's disease and depression[3].

Traditional bioelectronic systems rely on silicon-based digital-signal-processing hardware to detect and respond to neural activity[4]. Although they are highly advanced, these inorganic technologies face fundamental limitations in biocompatibility, flexibility and energy efficiency[4,5]. Silicon-based circuits are rigid and bulky, requiring careful isolation from biological tissue to prevent adverse reactions and mechanical stress. Additionally, their high power consumption and heat dissipation pose challenges for long-term implantation and

[1]Laboratory of Organic Electronics, Department of Science and Technology, Linköping University, Norrköping, Sweden. [2]Department of Electrical Engineering, Columbia University, New York, NY, USA. [3]Department of Neurobiology and Behavior, Cornell University, Ithaca, NY, USA. [4]Wallenberg Initiative Materials Science for Sustainability, Department of Science and Technology, Linköping University, Norrköping, Sweden. [5]Department of Chemistry, Chemistry Research Laboratory, University of Oxford, Oxford, UK. [6]Beijing National Laboratory for Molecular Sciences, Key Laboratory of Polymer Chemistry and Physics of the Ministry of Education, Center of Soft Matter Science and Engineering, College of Chemistry and Molecular Engineering, Peking University, Beijing, China. [7]Department of Pediatrics, University of California, Irvine, Irvine, CA, USA. [8]Andlinger Center for Energy and the Environment and Department of Electrical and Computer Engineering, Princeton University, Princeton, NJ, USA. [9]Department of Anatomy and Neurobiology, University of California, Irvine, Irvine, CA, USA. [10]Department of Electrical Engineering, University of California, Irvine, Irvine, CA, USA. [11]These authors contributed equally: Chi-Yuan Yang, Zifang Zhao, Han-Yan Wu. ✉e-mail: dion.kh@uci.edu; simone.fabiano@liu.se

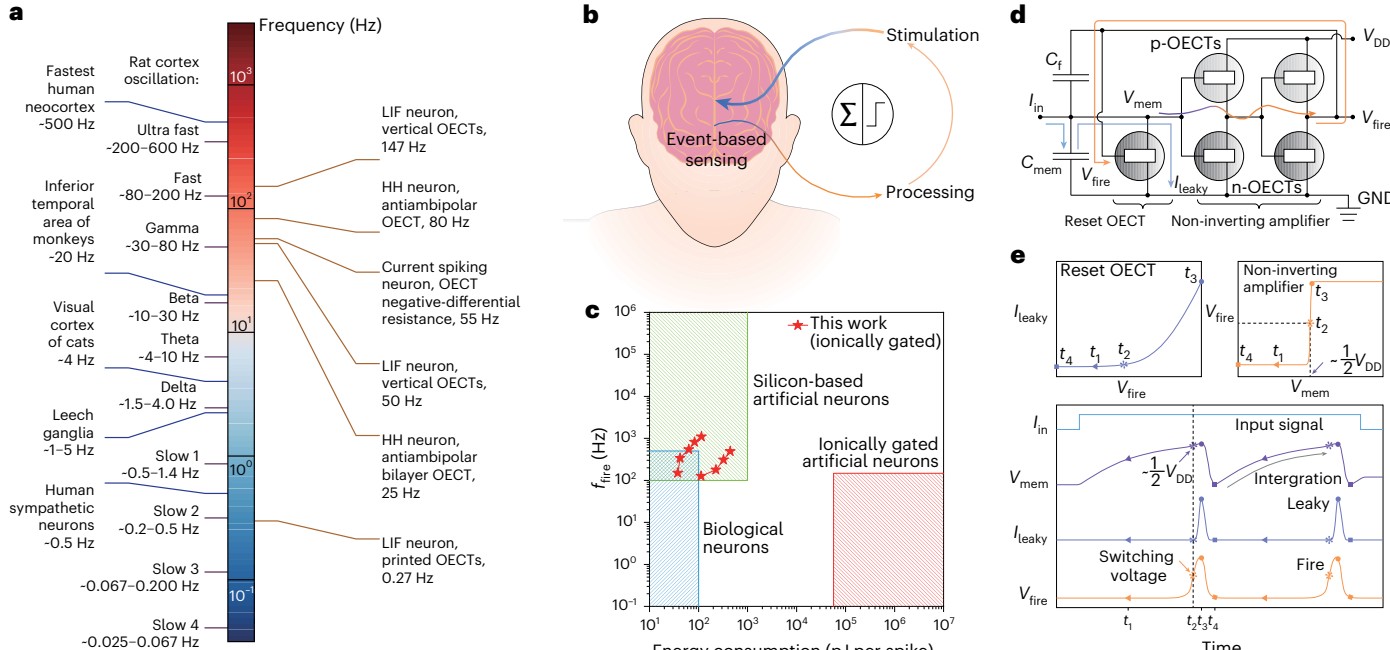

**Fig. 1 | OECNs and closed-loop neural stimulation. a**, Typical firing frequency range for biological neurons and ionically gated OECT-based artificial neurons. **b**, Schematic of the proposed closed-loop stimulation using OECNs. The symbols inside the circle represent signal summation (Σ, left) and thresholding (∫, right) functions. **c**, Energy consumption and firing frequency of biological neurons, state-of-the-art silicon-based artificial neurons and ionically gated artificial neurons. **d**, Circuit diagram of a LIF-type OECN. Orange, light blue, dark blue and purple represent $V_{fire}$, $I_{in}$, $I_{leaky}$ and $V_{mem}$, respectively. **e**, Schematic of the working principle of a LIF artificial neuron. HH denotes a Hodgkin-Huxley model neuron, and GND indicates ground.

continuous operation in energy-constrained environments[5]. Although miniaturization has improved integration, these systems still rely on complex architectures with rigid metallic enclosures, limiting their adaptability for next-generation bioelectronics[6].

To overcome these limitations, organic mixed ionic–electronic conductors (OMIECs)[7,8] have emerged as a promising alternative. Unlike traditional semiconductors, OMIECs support both ionic and electronic transport, enabling low-voltage operation, high transconductance and seamless biointegration[9–13]. Organic electrochemical transistors (OECTs), a key class of OMIEC devices, have demonstrated exceptional sensitivity to biological signals and are widely used in biosensors, neuromorphic computing and electrophysiology interfaces[14–16]. One of the most promising developments in OMIEC-based bioelectronics is the organic electrochemical neuron (OECN)[17], a class of spiking circuits that mimic biological neurons. OECNs leverage ion-mediated signal transduction to convert physiological inputs into action-potential-like outputs, making them ideal candidates for event-based sensing applications[18–22]. However, although recent developments in advanced materials and device architectures have improved OECN performance, achieving firing frequencies of up to ~140 Hz[23], existing OECNs still fall short of the biorealistic spiking frequencies required to interface with fast mammalian neurons and continue to suffer from excessive energy consumption and scalability challenges, limiting their potential for real-time neuromodulation.

The efficacy of OECNs in biomedical applications depends on their spiking frequency, energy efficiency and ability to provide closed-loop control. Slow-spiking OECNs (~1 Hz) can interface with the nerves of invertebrates (for example, the leech ventral nerve cord) but have limited applications[24]. Increasing the frequency to ~80 Hz enables event-based ion sensing for nerve stimulation in rodents[19,25]. However, mammalian neurons span a much broader range (0.5–1,000 Hz; Fig. 1a), making higher-frequency OECNs essential for applications such as epilepsy treatment[26] and Parkinson's disease management[27,28]. Additionally, although biological neurons consume ~100 pJ per spike[29], current OECNs operate at nanojoules or microjoules per spike. Reducing this

energy consumption to biorealistic levels is crucial to avoid excessive heat and noise. Finally, the real-time modulation of neurological disorders such as epilepsy requires continuous monitoring of abnormal neuronal activity and immediate intervention with deep brain stimulation upon detection.

In this Article, we report the development of a high-frequency, energy-efficient OECN-based sensor designed to overcome these limitations. By optimizing both materials and device architecture, we achieved event-driven sensors capable of responding within ~1 ms and operating across physiologically relevant frequencies up to 1.1 kHz with ultra-low energy demand (down to ~40 pJ per spike), matching the dynamic range of mammalian neurons. To demonstrate the functional capabilities of these ultra-fast, low-power sensors, we integrated them into a closed-loop neurostimulation system for real-time interventions. In a temporal lobe epilepsy model, the OECNs accurately detected interictal epileptiform discharges (IEDs) and delivered responsive stimulation to suppress pathological brain activity in vivo, combining speed, efficiency and adaptability. The findings highlight the potential of OECNs as a scalable, biocompatible and energy-efficient alternative to conventional silicon-based neuromorphic processors, paving the way for their use in next-generation bioelectronic medicine, neuromorphic computing and implantable neural interfaces.

## Design and implementation of the OECN's building blocks

OECNs, based on the leaky integrate-and-fire (LIF) model, incorporate a reset OECT and a non-inverting amplifier block comprising two cascaded OECT-based inverters (Fig. 1d). This architecture was chosen for its robustness and efficiency in hardware integration with OECTs. In this system, the input current ($I_{in}$) is temporally integrated on the membrane capacitor ($C_{mem}$), leading to a linear increase in membrane voltage ($V_{mem}$), as shown in Fig. 1e. Once $V_{mem}$ crosses the inverter's switching threshold, the amplifier triggers the generation of a voltage spike ($V_{fire}$) and activates positive feedback via the feedback capacitor ($C_f$). The rapid rise in $V_{fire}$ turns on the reset OECT,

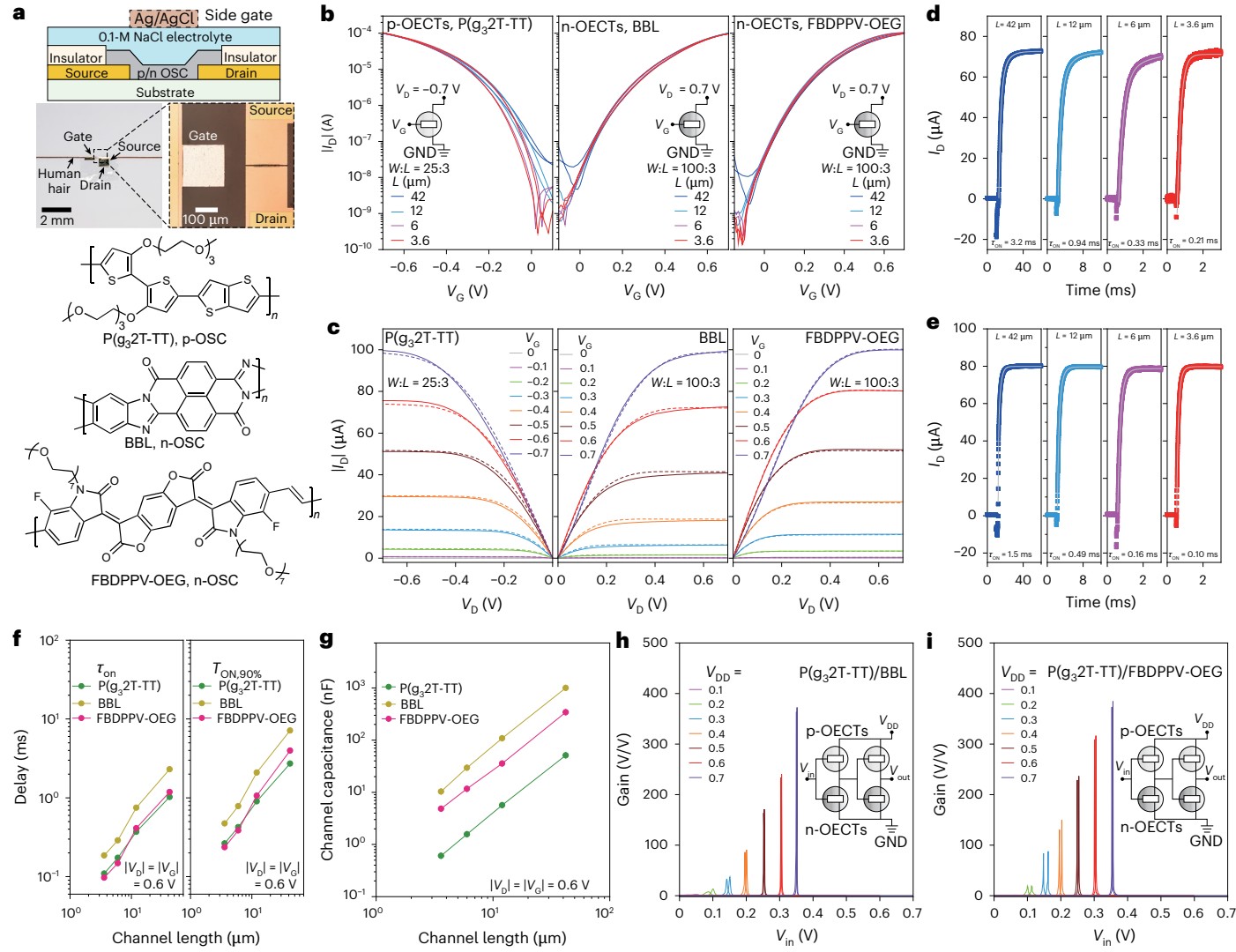

**Fig. 2 | Electrical characteristics of complementary OECT-based circuits.**
**a**, Cross-sectional view of the OECT (top) and chemical structures (bottom) of the p- (P(g₃2T-TT)) and n-type (BBL and FBDPPV-OEG) OMIECs. **b**, Transfer characteristics of p- and n-type OECTs with varying channel lengths ($L$). **c**, Typical output characteristics of p- and n-type OECTs ($L$ = 42 (dashed lines) or 3.6 μm

(solid lines)). **d,e**, Switching-on transient characteristics of n-type OECTs based on BBL (**d**) and FBDPPV-OEG (**e**). **f**, Summary of the rise-time parameters of the OECTs. **g**, Channel capacitances of the p- and n-type OECTs. **h,i**, Voltage gains of the OECN's two-stage inverter based on P(g₃2T-TT)/BBL (**h**) and P(g₃2T-TT)/FBDPPV-OEG (**i**). OSC denotes organic semiconductor.

discharging $C_{mem}$ and resetting $V_{mem}$ back to its baseline, completing the firing cycle (Fig. 1e and Extended Data Fig. 1 for SPICE simulations of the circuit). The spiking frequency can be tuned by adjusting $I_{in}$, $C_{mem}$ and $C_f$, but it is ultimately constrained by the propagation delay of the OECT-based inverters and the switching speed of the reset OECT. Therefore, achieving high-frequency OECNs requires fast OECT operation.

Since the cutoff frequency ($f_T$) of an OECT is directly proportional to the mobility of its channel material and inversely proportional to the square of its channel length ($f_T \propto \frac{\mu V_{DS}}{2\pi L^2}$, where μ is the carrier mobility and $V_{DS}$ is the drain-to-source voltage)[30], using high-mobility OMIECs and minimizing the OECT channel length enables the fabrication of fast and compact LIF OECNs (details in Methods and Supplementary Fig. 1a–c). The non-inverting amplifier block comprises two complementary inverters made from accumulation-mode OECTs[31], with the p-type OECT channel comprising the glycolated polythiophene P(g₃2T-TT) and the n-type OECT channel made of either poly(benzimidazo-benzophenanthroline) (BBL) or oligo(ethylene glycol)-substituted fluorinated benzodifurandione-based poly

(*p*-phenylene vinylene) (FBDPPV-OEG) (Fig. 2a). BBL and P(g₃2T-TT) were selected for their low threshold voltage, high stability and optimal OECT characteristics[32,33] (Supplementary Tables 1 and 2 and Supplementary Fig. 2), whereas FBDPPV-OEG was chosen for its high electron mobility and efficient charge transport properties (volumetric capacitance $C^*$ = 322 F cm⁻³, mobility $\mu$ = 0.12 cm² V⁻¹ s⁻¹ and $\mu C^*$ = 40 F cm⁻¹ V⁻¹ s⁻¹; Supplementary Fig. 3 and Extended Data Fig. 2a,b)[34]. The reset transistor is an n-type OECT made from either BBL or FBDPPV-OEG. The channel length was precisely defined via photolithography, with patterned gold source and drain electrodes determining the channel area. A thin Parylene C (PaC) layer controls the exposed electrode area (that is, parasitic capacitance), whereas a lateral Ag/AgCl gate electrode modulates the channel via an electrolyte (0.1 M NaCl aqueous solution). The OECTs were fabricated on flexible PaC substrates with a thickness of 5 μm (Fig. 2a).

We downscaled the OECT channel length while maintaining the width-to-length ($W:L$) ratio to ensure a constant channel resistance. The $W:L$ ratios were 100:3 for n-type OECTs and 25:3 for p-type OECTs, with the channel thicknesses optimized (Supplementary Fig. 4) to balance current levels and achieve symmetric performance. When reducing the

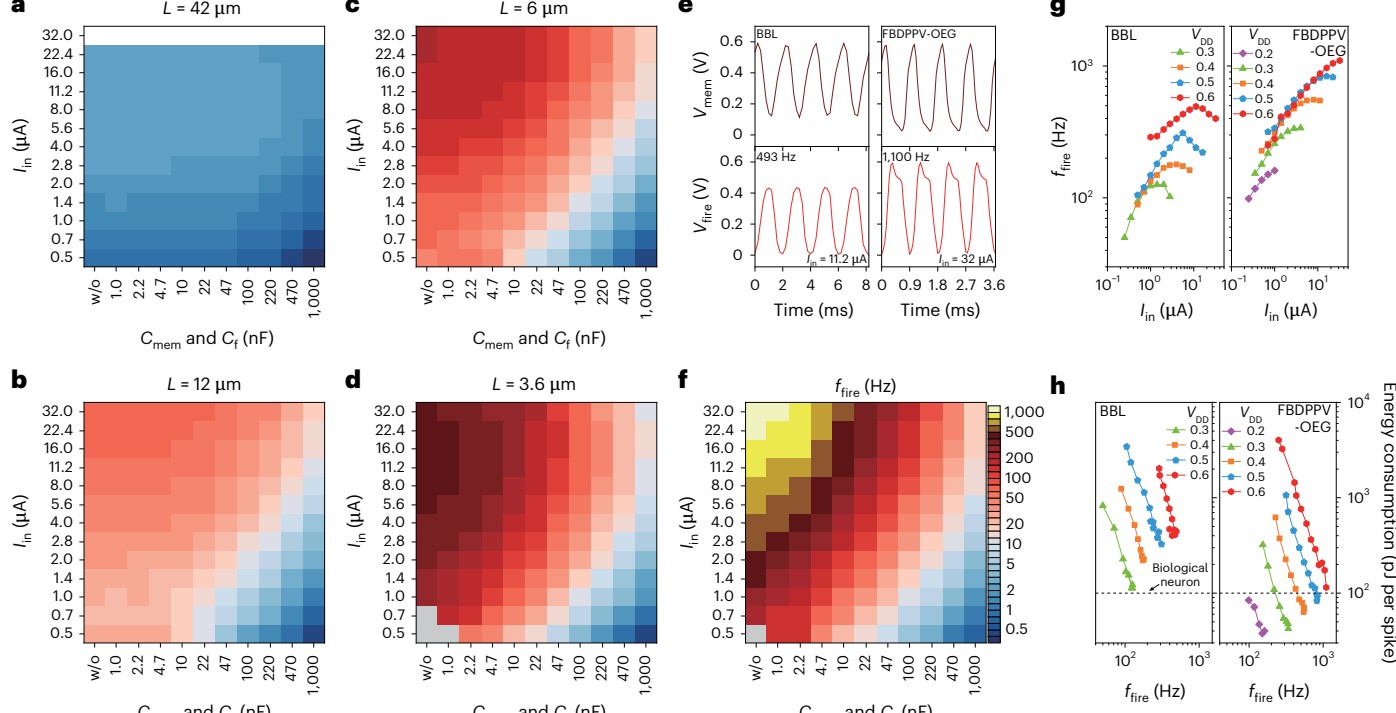

**Fig. 3 | Electrical characteristics of the OECNs. a–d,** Heatmaps showing the firing frequencies of P(g$_3$2T-TT)- and BBL-based OECNs as a function of input current, membrane capacitance and feedback capacitance for OECT channel lengths of 42 (**a**), 12 (**b**), 6 (**c**) and 3.6 μm (**d**) ($V_{DD}$ = 0.6 V for all). **e,** Voltage waveforms of action potential firing with $L$ = 3.6 μm and $V_{DD}$ = 0.6 V for P(g$_3$2T-TT)- and BBL-based (left) and P(g$_3$2T-TT)- and FBDPPV-OEG-based OECNs (right). **f,** Heatmap showing the firing frequencies of P(g$_3$2T-TT)- and FBDPPV-OEG-based OECNs as a function of input current, membrane capacitance and

feedback capacitance for $L$ = 3.6 μm ($V_{DD}$ = 0.6 V). **g,** Dependence of OECN firing frequencies on supply voltage and input current for P(g$_3$2T-TT)- and BBL-based (left) and P(g$_3$2T-TT)- and FBDPPV-OEG-based OECNs (right) (for both, without $C_{mem}$ and $C_f$; $L$ = 3.6 μm). **h,** Energy consumption per spike as a function of various supply voltages for P(g$_3$2T-TT)- and BBL-based (left) and P(g$_3$2T-TT)- and FBDPPV-OEG-based OECNs (right) (for both, without $C_{mem}$ and $C_f$; $L$ = 3.6 μm), with the dashed line marking the typical energy consumption of biological neurons[57]. w/o, without.

channel length from 42 to 3.6 μm, both p- and n-type OECTs retained their driving strength, exhibiting an on current of ~100 μA at a drain voltage $|V_D|$ of 0.7 V, as shown in Fig. 2b. To maintain a constant $W$:$L$ ratio, $W$ was proportionally reduced, ensuring that the current remained unchanged despite the decrease in $L$. Notably, no short-channel effects were observed at 3.6 μm, thanks to the bulk channel conductance, which prevented $V_D$-induced channel-length modulation. When the gate voltage $V_G$ was held constant and $|V_D|$ increased from 0 to 0.7 V, a clear transition from the linear region to the saturation region was observed in the output curves of P(g$_3$2T-TT), BBL and FBDPPV-OEG OECTs (Fig. 2c). In saturation mode, the drain current remained stable and independent of $V_D$, confirming robust OECT operation.

A narrow channel enables faster-switching OECTs, improving the speed of inverters and reset OECTs. When a $V_G$ pulse of 0.6 V is applied, P(g$_3$2T-TT) OECTs reach 90% of their saturation drain current ($T_{ON,90\%}$) within 3.8 ms for $L$ = 42, with a turn-on time constant ($\tau_{on}$) of 1.3 ms. Reducing the channel length to 3.6 μm substantially improves switching dynamics, decreasing $T_{ON,90\%}$ to 0.30 ms and $\tau_{on}$ to 0.12 ms (Extended Data Fig. 2c). For n-type BBL OECTs, $T_{ON,90\%}$ = 11 ms ($\tau_{on}$ = 3.2 ms) at $L$ = 42 μm, decreasing to $T_{ON,90\%}$ = 0.57 ms ($\tau_{on}$ = 0.21 ms) when $L$ is reduced to 3.6 μm (Fig. 2d). For n-type FBDPPV-OEG OECTs, $T_{ON,90\%}$ = 5.8 ms ($\tau_{on}$ = 1.5 ms) for $L$ = 42 μm, decreasing to $T_{ON,90\%}$ = 0.27 ms ($\tau_{on}$ = 0.10 ms) when $L$ decreases to 3.6 μm (Fig. 2e). These OECTs are at least four times faster than previously reported planar OECTs. The nearly linear correlation between operating frequency and channel length (Fig. 2f) suggests that further downscaling with higher-resolution photolithography techniques could push switching frequencies beyond 10 kHz (Extended Data Fig. 3a). To understand this improvement, we analysed the relationship between channel capacitance and channel length (Fig. 2g).

We found that the channel capacitance of the 3.6-μm OECT is two orders of magnitude smaller than that of the 42-μm OECT. Additionally, FBDPPV-OEG-based OECTs exhibit a 2.6-fold lower channel capacitance compared with those based on BBL, consistent with the material's lower $C^*$ (Supplementary Fig. 3), which reduces parasitic capacitance and results in approximately twofold faster operation (Extended Data Fig. 3). This reduction in capacitance minimizes the ionic charge needed to saturate the channel, substantially improving the switching speed.

We then developed the complementary non-inverting amplifier by cascading two inverters with 3.6-μm OECTs, which operate symmetrically in accumulation mode. Balanced driving strength was achieved by optimizing the P(g$_3$2T-TT), BBL and FBDPPV-OEG thicknesses to 5, 20 and 7 nm, respectively, ensuring efficient pull-up (p-type) and pull-down (n-type) operation. This configuration produced sharp voltage transfer characteristics with a well-defined state transition at $V_{in}$ = $V_{DD}/2$ (Fig. 2h,i and Supplementary Fig. 5), where $V_{in}$ and $V_{DD}$ are the input and supply voltage, respectively. The high symmetry of the two-stage inverters enables operation at a minimal $V_{DD}$ of 0.1 V, marking a substantial advancement in low-power OECT-based logic circuits. However, a $V_{DD}$ level above 0.5 V is preferred to minimize hysteresis. At $V_{DD}$ = 0.7 V, the DC voltage gain reached over 400 (V/V) (Fig. 2h,i). The two-stage complementary inverters exhibited exceptionally low power consumption, with a static power consumption ($P_{static}$) < 1 nW and a switching power consumption ($P_{switch}$) < 2 nW at $V_{DD}$ = 0.1 V. At $V_{DD}$ = 0.7 V, $P_{static}$ remained below 13 nW and $P_{switch}$ remained below 4 μW (Supplementary Fig. 5), demonstrating the efficiency of complementary inverters in minimizing power consumption. The delay of a single-stage inverter is estimated at 0.21 ms for $V_{DD}$ = 0.6 V (those of P(g$_3$2T-TT) and BBL are

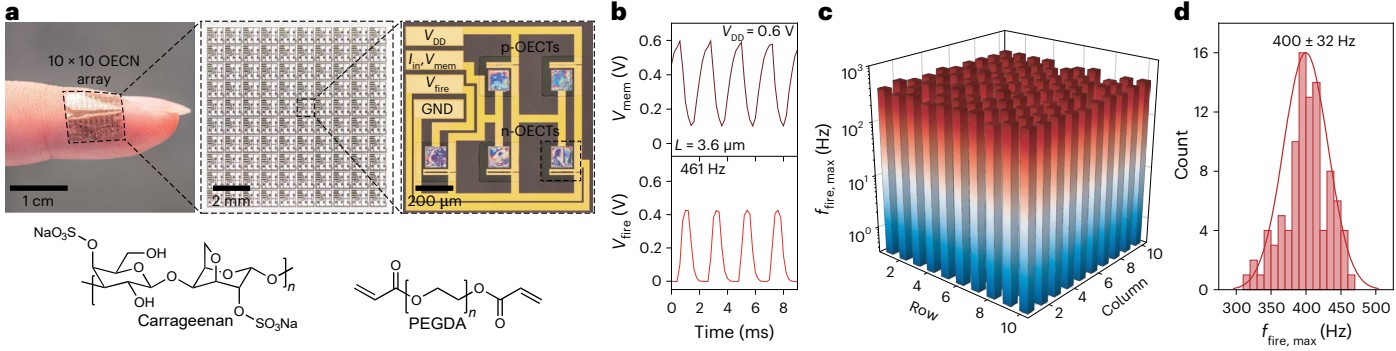

**Fig. 4 | Solidification and miniaturization of OECNs. a**, Top: a conformable solid-state 10 × 10 OECN array placed on a human finger (left), along with magnified microscopy images of the OECN. Bottom: chemical structures of parts of the solid-state electrolyte (made up from carrageenan, photo-crosslinked PEGDA and a 9:1 ratio of 0.1 M NaCl in water:glycerin) used to fabricate the 1 × 1 mm² OECN. **b**, Typical voltage waveforms of action potential firing of the solid-state P(g$_3$2T-TT)- and BBL-based OECN (without $C_{mem}$ or $C_f$; $I_{in}$ = 5.6 µA). **c**, Maximum firing frequencies in the 10 × 10 P(g$_3$2T-TT)- and BBL-based OECN array (without $C_{mem}$ or $C_f$; $I_{in}$ = 2.8–11.2 µA). **d**, Histograms showing the statistical distribution of the firing frequencies from 96 working BBL-based OECNs out of 100 in the array.

shown in Extended Data Figs. 4 and 5), over 10× faster than current state-of-the-art complementary inverters based on vertical OECTs (5.6 ms delay)[35].

## High-frequency OECNs with biological-level energy efficiency

Leveraging the fast-transient response of short-channel OECTs, we developed a series of high-speed OECNs by monolithically integrating the OECT-based amplifiers and the reset OECT. In this LIF OECN architecture, capacitors $C_{mem}$ and $C_f$ can be adjusted to minimize membrane capacitance and achieve high spiking frequency, allowing the inherent OECT gate-to-source capacitance to act as the membrane capacitance for current integration (Extended Data Figs. 1 and 2). The heatmaps in Fig. 3a–d offer a clear visual representation of the spiking frequency modulation for OECNs incorporating P(g$_3$2T-TT) and BBL-based OECTs. The intra-map colour scale illustrates how electrical parameters, including $I_{in}$, $C_{mem}$ and $C_f$, influence the OECN spiking behaviour, whereas the inter-map variation demonstrates how the OECT channel length effectively scales the spiking frequency of the LIF OECN, extending its operational range (Supplementary Figs. 6–9 and Extended Data Fig. 6). The diagonal trend in the heatmaps (from bottom right to top left) indicates that accelerating charge integration by increasing $I_{in}$ and reducing $C_{mem}$ enhances the OECN spiking frequency. At $L$ = 42 µm, the spiking frequency is limited to 2.3 Hz (Fig. 3a), even with optimized $I_{in}$ and $C_{mem}$, due to a prolonged inverter delay (~34 ms; Extended Data Fig. 4). Reducing the channel length to 12 and 6 µm increases spiking frequencies to 74 and 203 Hz, respectively, at $I_{in}$ = 32 µA and without $C_{mem}$ or $C_f$ (Fig. 3b,c). At $L$ = 3.6 µm, the spiking frequency surged to 493 Hz at $I_{in}$ = 11.2 µA and without $C_{mem}$ or $C_f$ (Fig. 3d,e). Notably, at this short channel length, OECNs maintain fast spiking even at $V_{DD}$ = 0.3 V (Supplementary Figs. 10–13) or remain sensitive to small input currents, achieving a spiking rate of 3 mHz at $I_{in}$ as low as 2 nA (Extended Data Fig. 6). To further enhance performance, we fabricated OECNs using the n-type FBDPPV-OEG-based OECTs for both the reset OECT and the pull-down OECTs in the two-stage amplifier, while keeping P(g$_3$2T-TT)-based OECTs unchanged. At $L$ = 3.6 µm, these OECNs achieved spiking frequencies up to 1,100 Hz and event-driven sensors capable of responding within ~1 ms (Supplementary Fig. 14 and Extended Data Fig. 7) at $I_{in}$ = 32 µA and without $C_{mem}$ or $C_f$ (Fig. 3e,f), highlighting the potential for ultra-fast OECNs. This frequency is nearly one order of magnitude higher than for previously reported OECNs[36].

With spiking frequencies of 0.3–1.1 kHz, these OECNs align with the firing rates of biological neurons in the primate neocortex[37]. Beyond extending the frequency range, downscaling the OECT geometry and reducing $V_{DD}$ substantially decrease OECN energy consumption, reaching 113 pJ per spike for BBL-based OECNs and 38 pJ per spike for

FBDPPV-OEG-based OECNs (Fig. 3h and Supplementary Figs. 10–13 and 15–19). This represents a four-orders-of-magnitude improvement over previous OECNs and achieves an energy efficiency comparable to those of biological neurons (~100 pJ per spike)[29]. Furthermore, the OECN footprint can be miniaturized to 1 × 1 mm² (Fig. 4a) using a solid-state electrolyte comprising a photo-patternable mixture of carrageenan, poly(ethylene glycol) diacrylate (PEGDA) and NaCl (Methods)[38]. These conformable OECNs, fabricated as a 10 × 10 array, achieve spiking frequencies of 400 ± 32 Hz, with a maximum firing frequency, $f_{fire}$ of 461 Hz (Fig. 4b–d).

## Responsive neural stimulation with OECNs

The ability to perform real-time, power-efficient computation on a biocompatible, conformable platform is crucial for responsive medical devices that detect pathological neural events with high accuracy. Electrical stimulation of neural circuits has shown promise in treating neurological disorders, including depression, epilepsy and pain management[39–41]. In epilepsy, targeted electrical stimulation during seizures can disrupt pathological activity and shorten seizure duration[42–44]. Outside of seizure events, fast detection and suppression of pathological coupling between IEDs and physiological cortical oscillations—particularly during non-rapid eye movement (NREM) sleep, when memory consolidation relies on precise hippocampal–cortical coordination[45–48]—can help to prevent the spread of epilepsy and mitigate memory impairments associated with abnormal neural activity[49,50]. In this regard, closed-loop electrical stimulation systems provide a promising approach for localized intervention at optimal moments without disrupting ongoing brain activity. These systems must detect specific electrophysiological waveforms (or biomarkers) in real time, deliver tailored stimulation and adapt to dynamic neural signals. With this in mind, we explored the potential of OECNs as a key component in developing closed-loop interventions for epilepsy. For experiments requiring long-term in vitro or in vivo operation, we employed BBL-based OECNs, which maintain stable, continuous operation for hours with minimal degradation, making them well suited for sustained neurostimulation applications (Extended Data Fig. 8).

First, we characterized the OECN's output in response to artificial hippocampal IED waveforms. IEDs are defined by fast, synchronized neuronal activities that generate large transients in local field potentials (LFPs), measured as the voltage difference between the recording and reference electrodes. The LFP signals were first amplified using a preamplifier and then converted into a current ($I_{in}$) for the OECNs using a 100-kΩ series resistor. Due to their fast operation, OECNs could generate multiple action potentials in response to IEDs that caused the $V_{mem}$ to cross the threshold, with a stable firing frequency of ~300 Hz (Fig. 5a,b and Extended Data Fig. 9a–c). The firing rate of OECNs can

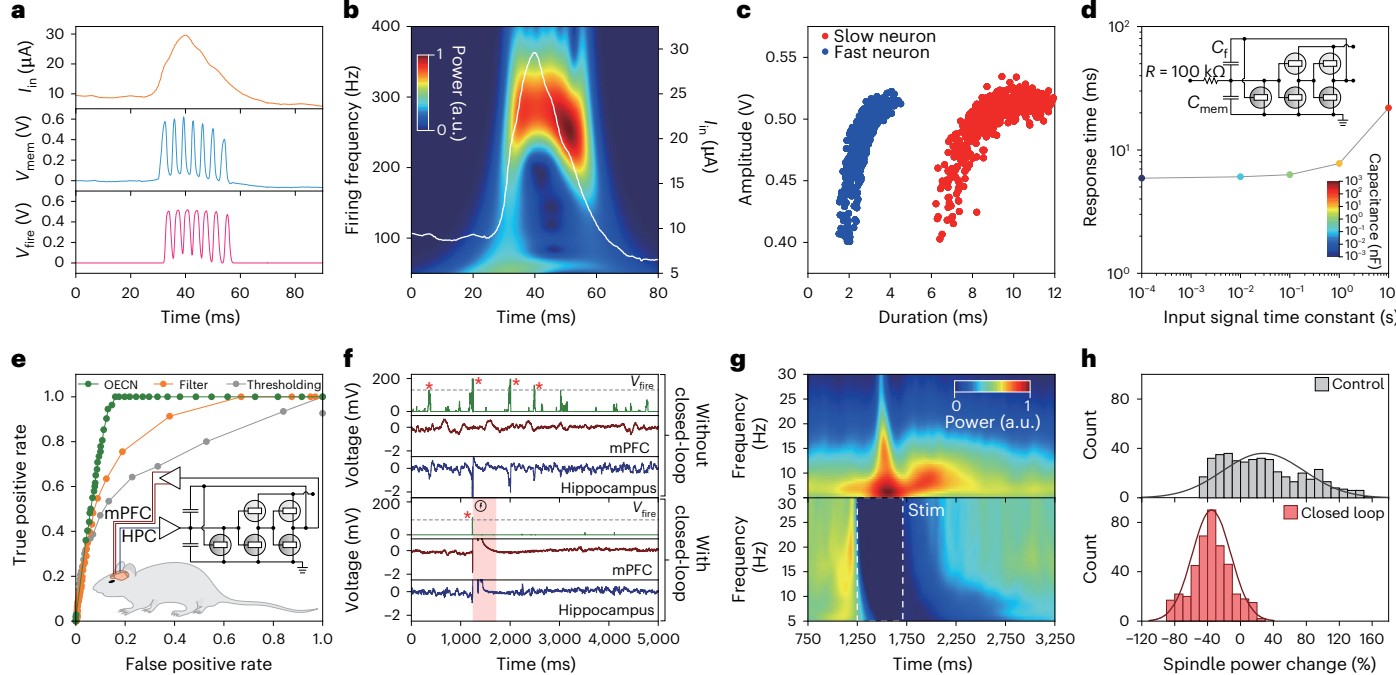

**Fig. 5 | OECN-enabled neural intervention for temporal lobe epilepsy.**

**a**, Sample waveforms of an OECN ($V_{mem}$ and $V_{fire}$) in response to a hippocampal epileptiform discharge ($I_{in}$). **b**, Time–frequency spectrogram of $V_{fire}$, showing the frequency response of OECN towards an IED. The input signal ($I_{in}$) is over-imposed as a white curve. **c**, Comparison of the IED stimulation response amplitudes of 400- (blue; fast OECN; $n_{IED}$ = 717) and 71-Hz OECNs (red; slow OECN; $n_{IED}$ = 751). **d**, OECN response time as a function of input-stage time constant. The colour gradient represents different capacitance values, with colours ranging from blue to red representing a capacitance series of 1 pF, 100 pF, 1 nF, 10 nF, 100 nF and 1 μF. Inset: circuit diagram. **e**, Receiver operating characteristics curve demonstrating the accurate IED detection of an OECN, compared with a filter-based detector and a simple voltage threshold detector ($n_{IED}$ = 380). Inset: illustration of the experimental setup, showing the OECN input from hippocampal area CA1 (HPC) and stimulation delivered to a pair of bipolar stimulating electrodes in the mPFC

with an OECN-controlled stimulation isolator. **f**, Suppression of pathological IED–spindle coupling with OECN-enabled closed-loop stimulation. Sample raw traces show hippocampal (blue) and mPFC signals (red) during NREM sleep, without (top) and with (bottom) closed-loop stimulation (red shading, with the beginning of stimulation indicated by a lightning bolt symbol). IEDs are marked with red asterisks. Plots of $V_{fire}$ without (top) and with (bottom) closed-loop stimulation are shown in green. The grey dashed lines indicate the $V_{fire}$-based voltage threshold used to detect IEDs and, in the closed-loop condition, to trigger stimulation. **g**, Sample time–frequency spectrograms of IED-triggered responses in the mPFC without closed-loop control (top; $n_{IED}$ = 423) and with an effective closed-loop spindle suppression protocol (bottom; $n_{IED}$ = 427). Stimulation and the amplifier recovery period are highlighted by a white dashed box labelled Stim. **h**, Statistics of the spindle band power change following IEDs, either with closed-loop stimulation (red) or without stimulation (grey).

be tuned to control the number of spikes per IED by adjusting the capacitance. Faster-spiking neurons (400 Hz) enable earlier IED detection with a shorter active period (2.81 ± 0.021 ms; $n$ = 717), making them suitable for high-frequency stimulation protocols. In contrast, slower neurons (71 Hz) maintain a longer active period (9.37 ± 0.045 ms; $n$ = 751), which is effective for large-scale stimulation protocols (Fig. 5c and Extended Data Fig. 9d). To further assess the detection onset time of these fast OECNs, we varied the capacitance of $C_{mem}$ from 1 pF to 10 μF. We found that when $C_{mem}$ was below 1 nF, the delay between IED onset and OECN firing remained stable at ~6 ms, aligning with the time constant of the input stage (Fig. 5d). This demonstrates the capability of OECNs to detect neural oscillations within the first cycle, in stark contrast with conventional filtering and thresholding methods, which can only detect oscillations after the second period, substantially hindering real-time interventions[51].

Upon confirming the ability of OECNs to generate the required IED stimulation response pattern, we deployed them for responsive in vivo neural interventions. We implanted microelectrodes in the medial prefrontal cortex (mPFC) and the CA1 area of the dorsal hippocampus in rats with focal epilepsy induced using an electrical kindling protocol (Methods). Real-time hippocampal electrophysiological recordings were fed into an OECN-based circuit for in vivo IED detection. To assess the detection accuracy, we compared the OECN with a digital-signal-processing-based detection method that employed filtering, rectification, power enveloping and thresholding using a silicon-based embedded system. The OECNs exhibited substantially higher detection

accuracy while consuming far less power than the other approaches (Fig. 5e and Supplementary Fig. 20).

After confirming the effectiveness of OECNs as biomarker detectors, we developed an OECN-controlled responsive device to suppress IED-induced disruptions in cortical neural activity. Our previous work demonstrated that a slow-varying Gaussian waveform effectively counteracts a pathological downstate in the mPFC, preserving cortical function[5]. Notably, the OECN output can be low-pass filtered to generate this waveform (Fig. 5f and Extended Data Fig. 9e). We then evaluated the efficacy of OECN in protecting the cortex by analysing the mPFC LFP power spectrum following hippocampal IEDs. The results showed a substantial reduction in spindle band power (10–15 Hz) after responsive stimulation, indicating that the OECN-based system successfully blocked pathological activity in the mPFC (Fig. 5g,h). Importantly, the OECN output remained stable throughout the entire 90-min recording session (Extended Data Fig. 9f). In addition, we tested whether OECNs could respond to seizure onset, characterized by a high occurrence of population spikes similar to IEDs. The OECNs successfully detected ictal activity during epileptic seizures (Extended Data Fig. 9g, Supplementary Fig. 20 and Supplementary Table 3).

## Conclusion

We have demonstrated ultra-fast (up to 1.1 kHz) and energy-efficient (down to 38 pJ per spike) OECNs capable of closed-loop neural stimulation. By optimizing the materials and device architecture, we developed conformable, biocompatible, event-based sensors that operate at

physiologically relevant frequencies with low energy consumption. In vitro experiments showed that OECNs respond effectively to electrophysiological signals from an epilepsy model, maintaining stable oscillation frequencies during IEDs and accurately detecting pathological events. Enabled by high-gain, nonlinear amplification, they therefore outperform conventional detectors. In vivo, OECNs detected hippocampal IEDs in real time and triggered responsive stimulation to suppress pathological spindle activity in the mPFC, highlighting their potential for clinical application. Our results demonstrate the distinct advantages of OECNs over traditional silicon-based electronics, including direct analogue signal processing (without the need for analogue-to-digital conversion), high accuracy and selectivity, low operating voltage, mechanical flexibility and compatibility with long-term implantation. The use of clinically approved PaC substrates[52] and operation at voltages below the hydrolysis limit further enhance the biocompatibility and stability for chronic use.

Although OECNs hold great promise for clinical translation, several challenges remain. Clinical recordings are often affected by electromyographic contamination[53]. In our experiments, electromyographic signals remained below the IED detection threshold and did not substantially impact performance. However, established strategies, such as noise detection[54], median removal[55], upper thresholding[56] or a combination of these in more complex scenarios, can further improve the detection accuracy, and OECNs are fully compatible with these approaches.

In addition, although the planar OECT configuration used here enables ease of integration in complementary circuits and stable, cross-talk-free operation, its transient response is slower than that of vertical and/or internal-gated designs[23,25]. Future performance improvements could leverage these alternative architectures, which offer shorter channel lengths and higher speed, and represent promising directions for advancing OECN performance. Overall, the conformable and versatile nature of OECNs, combined with their high performance, low energy consumption and compatibility with clinical materials and fabrication methods, positions them as strong candidates for next-generation implantable devices in bioelectronic medicine and brain–computer interfaces.

## Methods

All of the animal experiments were approved by the Institutional Animal Care and Use Committee of Columbia University (Assurance ID: D16-00003).

### Materials

BBL[32], FBDPPV-OEG[34] and the glycolated polythiophene (P(g₃2T-TT))[3] were synthesized according to previous reports. PEGDA, 2-hydroxy-4′-(2-hydroxyethoxy)-2-methylpropiophenone, carrageenan, glycerin, 3-(trimethoxysilyl)propyl methacrylate, 1,1,2,2-tetrachloroethane, hexafluoroisopropanol and methanesulfonic acid were purchased from Sigma–Aldrich.

### Fabrication of OECTs and OECNs

The fabrication steps are reported in Supplementary Fig. 1. For the conformable substrate, standard microscope glass slides were cleaned via successive sonication in acetone, deionized water and isopropyl alcohol and then dried in nitrogen. The industrial surfactant Micro-90 (2%) was spin-coated as an anti-adhesive layer, and a 5-μm-thick PaC layer was deposited on the glass slides. For the electrodes and interconnects, 5-nm-thick Cr and 50-nm-thick Au layers were thermally deposited on the PaC layer and photolithographically patterned by wet etching. A 1-μm-thick layer of PaC was then deposited, along with 50 μl 3-(trimethoxysilyl)propyl methacrylate to enhance adhesion, serving as an insulating layer to prevent capacitive effects at the metal–liquid interface. For the gate, an anti-adhesive layer of Micro-90 (2%) was spin-coated, followed by the deposition of a 2-μm-thick PaC layer as a sacrificial layer.

A 5-μm-thick positive photoresist (AZ10XT520CP) was spin-coated on the PaC layer, followed by photolithographic patterning and development using AZ Developer. Afterward, plasma reactive-ion etching (150 W; $O_2$ = 500 sccm; $CF_4$ = 100 sccm; 360 s) was performed to define the gate region. A 100-nm-thick Ag layer was deposited onto the PaC layer by thermal evaporation and then converted to Ag/AgCl by immersion in 1 mM HCl aqueous solution for 1 min. The Ag/AgCl gate was further patterned by lift-off, achieved by peeling off the sacrificial PaC layer.

For the n-type OECTs, similar to the gate patterning steps, a 2-μm-thick PaC layer was deposited as a sacrificial layer for the n-type channel. The channel region was defined by photolithographic patterning using AZ10XT520CP, followed by development and reactive ion etching under similar conditions to those described above. A 20-nm-thick BBL layer (or 7-nm-thick FBDPPV-OEG layer) was then deposited by spin-coating the BBL in a methanesulfonic acid solution of 2.5 mg ml⁻¹ (or the FBDPPV-OEG in a hexafluoroisopropanol solution of 3 mg ml⁻¹) onto the PaC layer, followed by water immersion and drying in nitrogen. Finally, BBL or FBDPPV-OEG was patterned by sacrificial PaC lift-off. For the p-type OECTs, we followed a process identical to that described above for the patterning of n-type OECTs. A 5-nm-thick P(g₃2T-TT) layer was deposited by spin-coating a P(g₃2T-TT) solution (1,1,2,2-tetrachloroethane; 3 mg ml⁻¹) and patterned by sacrificial PaC lift-off.

For the electrolyte, for regular OECTs and OECNs with 0.1 M NaCl aqueous solution, a polydimethylsiloxane well was attached to the device and the aqueous electrolyte was added. For the solid electrolyte, 1.5 g PEGDA, 0.4 g 2-hydroxy-4′-(2-hydroxyethoxy)-2-methylpropiophenone, 0.18 g carrageenan and 0.3 g glycerin were added to 10 g water and stirred at 110 °C. The solution was spin-coated onto the top of the device to form a 1-μm-thick transparent film. A 390-nm ultraviolet exposure of 9,000 mJ cm⁻² was applied to crosslink the solid electrolyte region. Then, the solid electrolyte was developed in water, immersed in a 0.1-M NaCl solution in a 1:9 glycerin:water mixture for 30 min and dried gently under a nitrogen flow. In the solid electrolyte, the photo-crosslinked carrageenan–PEGDA network provides a hydrated skeleton, whereas glycerin can substantially reduce water evaporation. Additionally, NaCl provides the necessary ions for the OECT or OECN. Even if the water in the solid electrolyte evaporates after being stored for a long time, re-immersing it in a 0.1-M NaCl solution in a glycerin:water mixture (1:9) and drying with a nitrogen flow can fully restore its performance. Finally, the edge of the glass slide was cut to allow the device to be peeled off as a whole. The channel width-to-length ratio was $W:L$ = 25:3 for p-OECTs and 100:3 for n-OECTs, for all channel lengths ($L$ = 3.6, 6.0, 12.0 and 42.0 μm).

### Electrical characterization of OECTs and OECNs

The OECTs were characterized by Keithley 4200A-SCS (with the modules 4225-PMU Ultra Fast I-V Module and 4225-RPM Remote Amplifier/Switch). The OECNs were characterized by Keithley 4200A-SCS (to supply $V_{DD}$ and to record waveforms below 5 Hz) and Agilent Infiniium 54832D Oscilloscope (to record waveforms above 0.3 Hz; internal resistance = 1 MΩ).

### Animal surgical procedure

A total of four male Long-Evans rats (Charles River Laboratories; 3–6 months old) were implanted with probes inserted into the mPFC and/or hippocampus. No animals were excluded from the study. A bipolar stimulating electrode based on tungsten wires was implanted in the hippocampal commissure. Two miniature stainless steel screws were fixed to the skull above the cerebellum to serve as ground and reference electrodes. Rats were kept on a regular 12 h light/12 h dark cycle and housed in pairs before implantation, but separated afterward. No previous experimentation had been performed on these rats. The animals were initially anaesthetized with 2% isoflurane and maintained under anaesthesia with 0.75–1.00% isoflurane during the surgery. Craniotomy was performed at three different sites:

the mPFC (anterior–posterior = 3.5 mm; medial–lateral = −0.8 mm; dorsal–ventral = 2.4 mm), hippocampus (anterior–posterior = −3.5 mm; medial–lateral = 3.0 mm; dorsal–ventral = 2.0 mm) and commissure (anterior–posterior = −0.5 mm; medial–lateral = 0.8 mm; dorsal–ventral = 4.2 mm). Dura mater was removed individually, and the probes were inserted to the target areas. The craniotomies were covered with gel-foam and sealed using a silicone elastomer.

### Validation of electrophysiological recordings

In vivo recordings were manually classified into wake, rapid eye movement and NREM epochs based on the theta/delta ratio, with additional movement information extracted from the onboard accelerometer[1]. Offline epileptiform discharge detection has been described previously[2].

### Epileptic animal modelling

Intracranial brain stimulation was delivered to the hippocampal commissure and mPFC during NREM sleep using bipolar electrodes (50-μm-diameter tungsten-polyimide insulated wires) and a stimulation isolator (STG4002; Multi Channel Systems). The electrical stimulation was used to induce hippocampal seizures using a kindling protocol (60-Hz, bipolar, 1-ms pulses delivered to the hippocampus commissure for 2 s). The amplitude of intracranial stimulation was derived empirically by titration of the stimulation amplitude to the network effect.

### In vitro validation

Previously recorded epileptic activity from three rats was streamed to the OECN via a digital-to-analogue converter (PCIe-6321; National Instruments). The signal was level shifted and scaled to match the OECN's input range (0–0.6 V) using a signal processor (Moku:Go; Liquid Instruments). The OECN was placed under a probe station and powered by a DC power supply. The $V_{fire}$ and $V_{mem}$ signals were recorded using an analogue-to-digital converter (PCIe-6321; National Instruments).

### Closed-loop stimulation

The animal was placed in its home cage, then electrodes were connected to a head stage (RHD2000; Intan Technologies). The amplified neural signal was level shifted and scaled to 0–0.6 V using a signal processor (Moku:Go; Liquid Instruments) before being applied to the OECN's $V_{mem}$. The OECN was connected under a probe station and powered by a DC power supply. The $V_{fire}$ and $V_{mem}$ signals were level shifted to 1.5 V and recorded using an analogue-to-digital converter (RHD2000; Intan Technologies). The 0–0.6 V $V_{fire}$ signal was scaled to 0–3.3 V by the signal processor (Moko:Go; Liquid Instruments) and used to control stimulation via the STG4002 (Multi Channel Systems). When stimulation was triggered, a pre-registered 1-V, 200-ms Gaussian waveform was delivered. A detailed schematic of the signal flow is shown in Extended Data Fig. 10.

### Reporting summary

Further information on research design is available in the Nature Portfolio Reporting Summary linked to this article.

### Data availability

The data supporting the findings of this study are available within the paper and its Supplementary Information files. Source data are provided with this paper.

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

## Acknowledgements

This work was financially supported by the Knut and Alice Wallenberg Foundation (grant 2021.0058, supporting research within the Wallenberg Wood Science Center and Wallenberg Initiative Materials Science for Sustainability), Swedish Research Council (2020-03243, 2022-04053, 2022-04553 and 2024-04871), European Research Council through the ERC Consolidator Grant project INFER (101125879), European Commission through the FET-OPEN project MITICS (964677), the Pathfinder OPEN project ICONIC (101129638) and the MSCA-IF-2020 project S-OECN (101152690), Swedish Foundation for Strategic Research (IS24-0162), VINNOVA (2023-01337) and the Swedish Government Strategic Research Area in Materials Science on Functional Materials at Linköping University (Faculty Grant SFO-Mat-LiU 2009-00971). J.P. acknowledges financial support from the National Natural Science Foundation of China (grants 22020102001 and 22335002).

## Author contributions

D.K. and S.F. conceived the idea and designed the project. C.-Y.Y. and H.-Y.W. synthesized BBL, fabricated the OECNs and conducted the electrical characterization. J.-D.H., T.L. and P.C.H. assisted with electrical characterization of the OECNs. J.J. and M.X. fabricated the solid-state OECNs. M.M. helped with the device fabrication. Z.Z., S.S., J.G. and D.K. performed the in vitro and in vivo tests. A.M. and I.M. synthesized P(g₃2T-TT). X.-Y.W. and J.P. synthesized FBDPPV-OEG. M.B. contributed to the scientific discussions. C.-Y.Y., Z.Z., D.G., D.T., D.K. and S.F. wrote the paper. All authors contributed to discussions and paper preparation.

## Funding

## Competing interests

The authors declare no competing interests.

## Additional information

**Extended data** is available for this paper at

**Correspondence and requests for materials** should be addressed to Dion Khodagholy or Simone Fabiano.

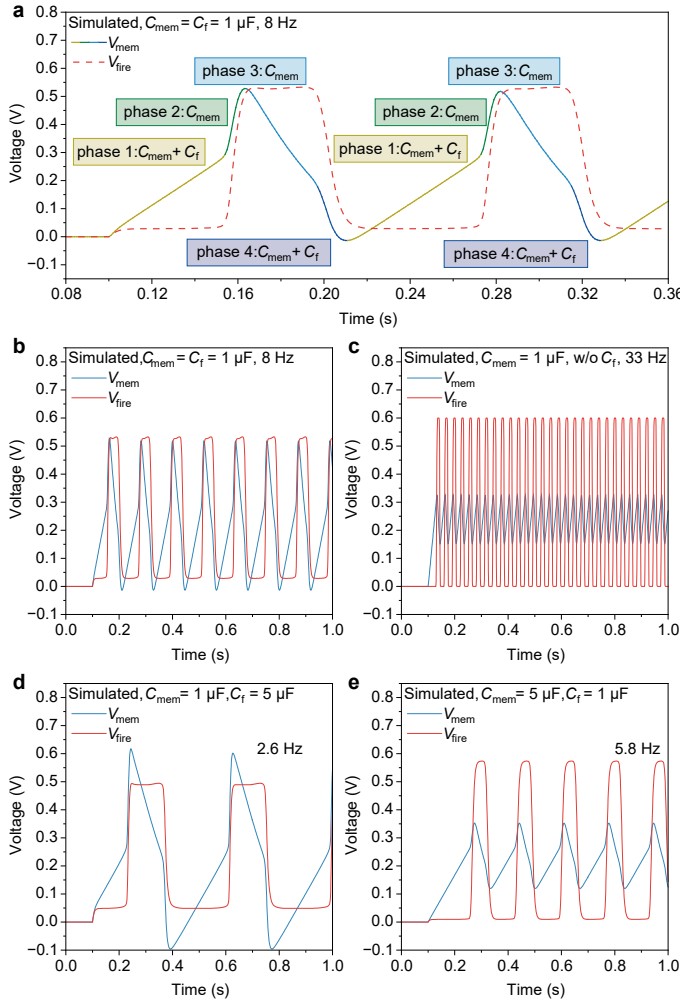

**Extended Data Fig. 1 | SPICE simulation of LIF-OECN spiking behavior.**
**a**, Simulated charging and discharging dynamics of $V_{mem}$. These SPICE simulations illustrate how $C_{mem}$ and $C_f$ influence the spiking dynamics, summarized as follows: (Phase 1) At the beginning of the spiking cycle, $C_{mem}$ and $C_f$ operate in parallel to integrate the input charge; (Phase 2) When the threshold is reached, $V_{fire}$ switches HIGH, and $V_{mem}$ rises sharply as $C_{mem}$ is charged by both $I_{in}$ and feedback voltage through $C_f$; (Phase 3) The HIGH $V_{fire}$ turns on the reset OECT, which discharges $C_{mem}$; (Phase 4) When $V_{mem}$ drops below the threshold, $V_{fire}$ switches LOW, and $V_{mem}$ rapidly discharges through both $C_{mem}$ and $C_f$. **b-c**, Comparison of LIF neuron spiking (**b**) with and (**c**) without $C_f$. **d-e**, Comparison of LIF neuron spiking with different $C_{mem}/C_f$ ratio: (**d**) 1:5 and (**e**) 5:1.

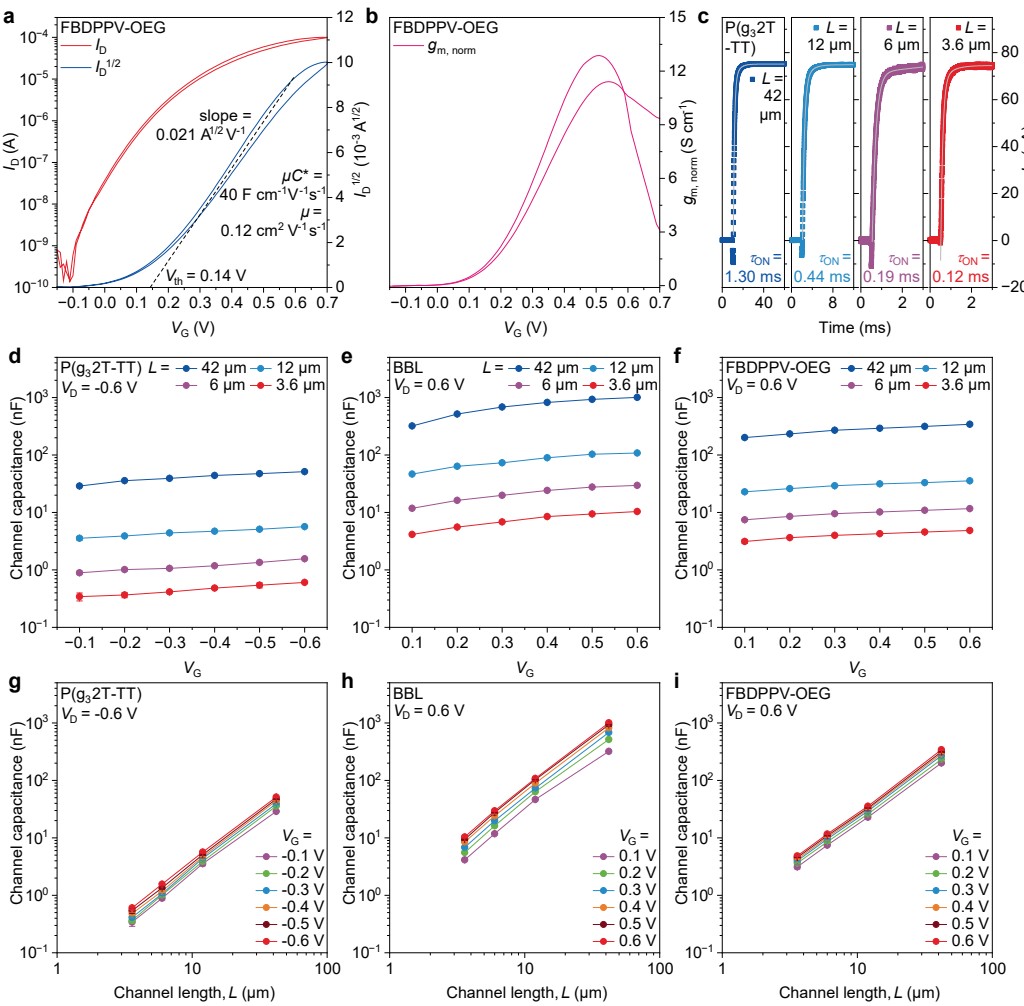

**Extended Data Fig. 2 | OECT performance. a**, Transfer curves of FBDPPV-OEG based OECT ($L$ = 3.6 μm). **b**, Geometry-normalized transconductance ($g_{m,norm}$) of FBDPPV-OEG based OECT ($L$ = 3.6 μm). **c**, Switching-on transient characteristics of the p-type OECTs based on P($g_3$2T-TT). **d-f**, Channel capacitance of P($g_3$2T-TT)-based OECTs (**d**), BBL-based OECTs (**e**), and FBDPPV-OEG based OECTs (**f**) under different $V_G$. **g-i**, Channel capacitance of P($g_3$2T-TT)-based OECTs (**g**), BBL-based OECTs (**h**), and FBDPPV-OEG based OECTs (**i**) within various channel lengths. Error bars represent the standard deviation. n = 5 samples.

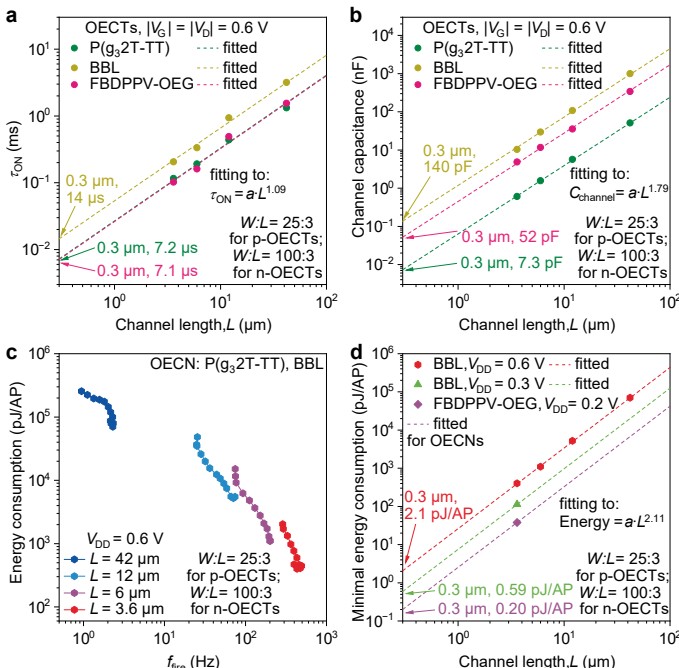

**Extended Data Fig. 3 | Downscaling extrapolation. a**, Downscaling extrapolation of switching-on transient characteristics of the OECTs. **b**, Downscaling extrapolation of channel capacitance of the OECTs.

**c**, Energy consumption of the OECNs based on P(g₃2T-TT)/BBL within various channel lengths. **d**, Downscaling extrapolation of energy consumption of the OECNs based on both P(g₃2T-TT)/BBL and P(g₃2T-TT)/FBDPPV-OEG.

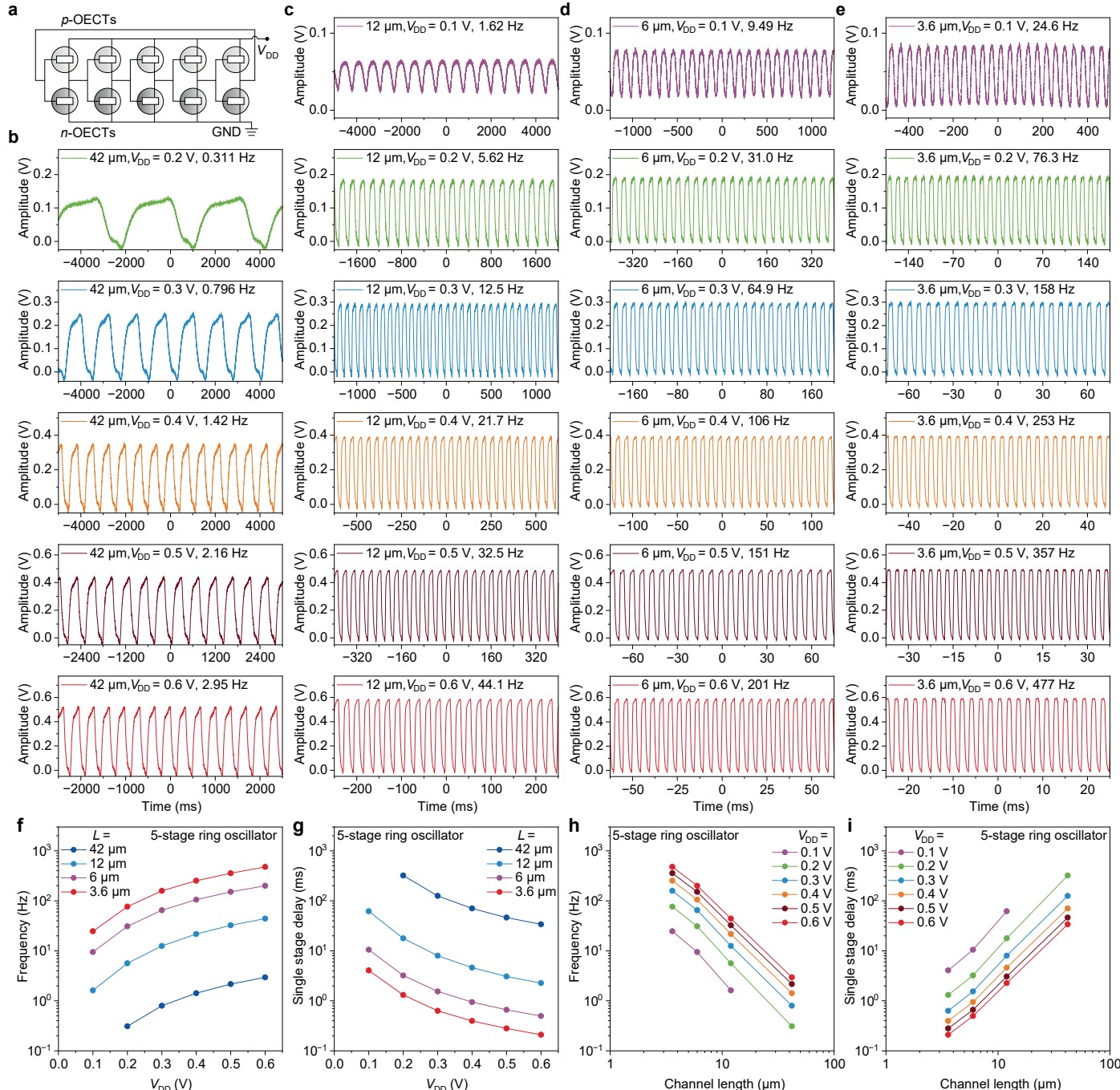

**Extended Data Fig. 4 | Ring oscillators. a**, Schematic of a 5-stage ring oscillator circuit based on P(g₃2T-TT)/BBL. **b-e**, Output characteristics of the 5-stage ring oscillator (based on P(g₃2T-TT)/BBL) within different channel lengths: $L$ = 42 μm (**b**), 12 μm (**c**), 6 μm (**d**), and 3.6 μm (**e**) at $V_{DD}$ = 0.1 to 0.6 V.

**f-g**, Output frequency (**f**) and single-stage delay (**g**) of the 5-stage ring oscillator at various $V_{DD}$. **h-i**, Output frequency (**h**) and single-stage delay (**i**) of the 5-stage ring oscillator with various channel lengths.

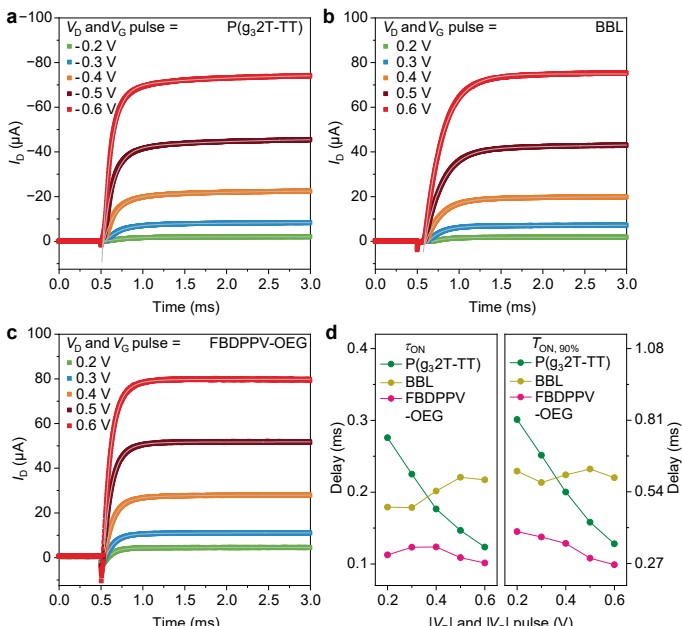

**Extended Data Fig. 5 | OECT transient response. a-c,** Switching-on transient characteristics of OECTs based on (**a**) P(g₃2T-TT), (**b**) BBL, and (**c**) FBDPPV-OEG at different $V_D$ biases and $V_G$ pulses. (**d**) Summary of the OECTs' voltage-dependent transient response. As the voltage decreases, the switching-on transient response becomes approximately 2.7× slower for P(g₃2T-TT) and less than 1.3× slower for FBDPPV-OEG, while it remains unchanged for BBL. This results in longer delays in the complementary circuit at lower voltages.

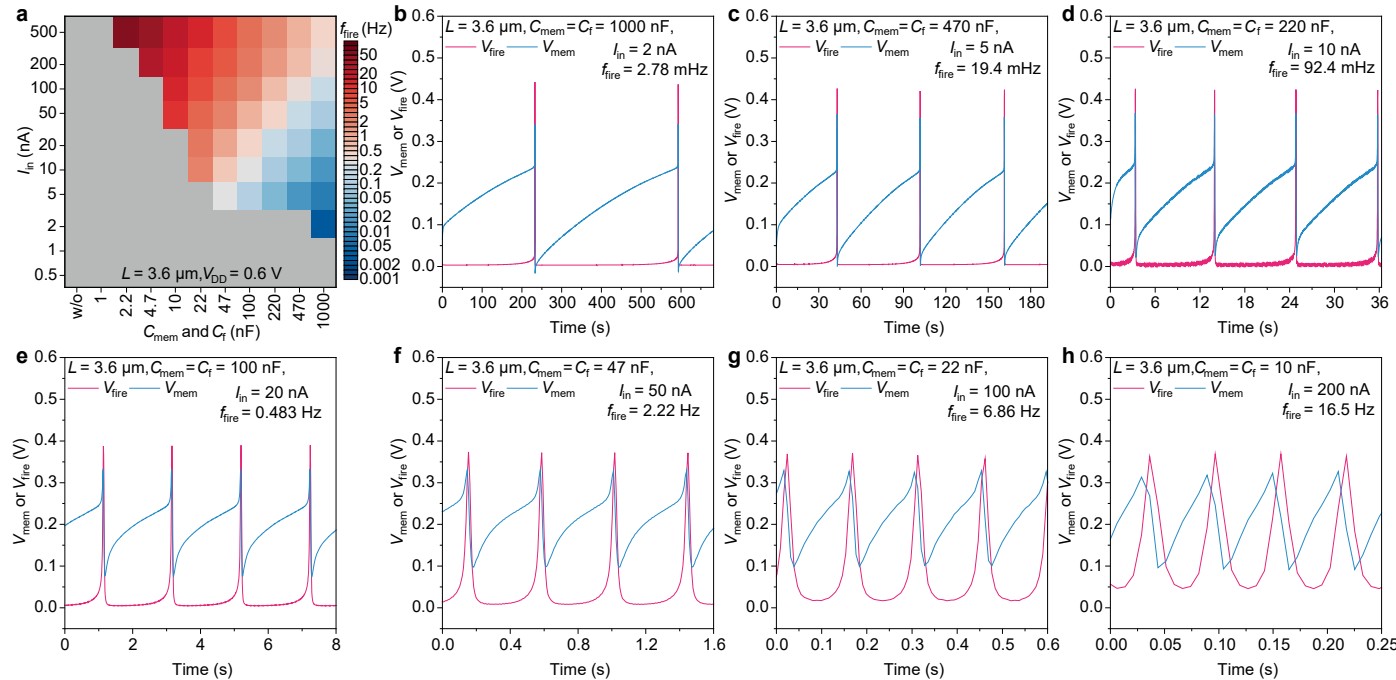

**Extended Data Fig. 6 | OECN spiking performance. a**, Heat map showing action potential firing frequencies as a function of input currents and membrane capacitances (based on P(g$_3$2T-TT)/BBL, $L$ = 3.6 μm and $V_{DD}$ = 0.6 V), with a detection limit of $I_{in}$ = 2 nA. Grey squares indicate conditions where no spikes were generated. **b-h**, Spiking behavior under various input currents and capacitances: 2 nA/1,000 nF (**b**), 5 nA/470 nF (**c**), 10 nA/220 nF (**d**), 20 nA/100 nF (**e**), 50 nA/47 nF (**f**), 100 nA/22 nF (**g**) and 200 nA/10 nF (**h**).

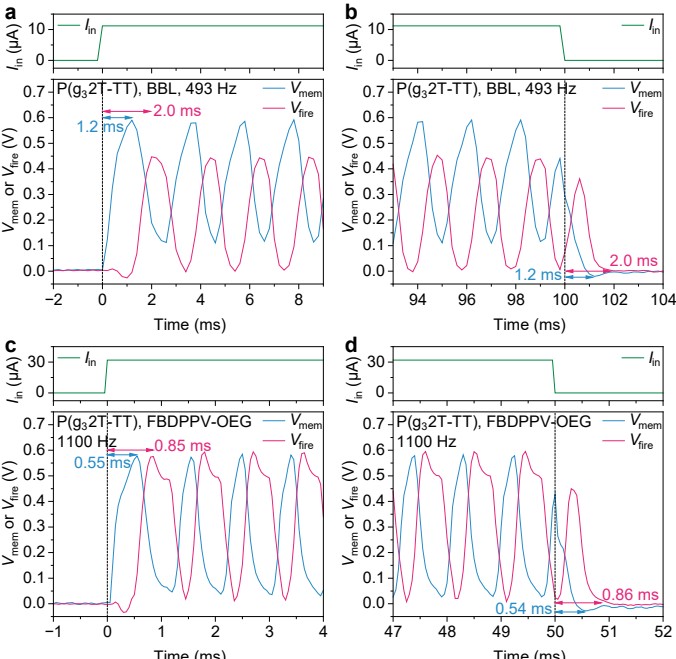

**Extended Data Fig. 7 | Event-driven sensing. a-b**, Transient response of OECNs based on P(g₃2T-TT)/BBL ($L$ = 3.6 μm, without $C_{mem}$ and $C_f$, $V_{DD}$ = 0.6 V), showing a response time of less than 2 ms: input signal on (**a**) and input signal off (**b**).

**c-d**, Transient response of OECNs based on P(g₃2T-TT)/FBDPPV-OEG ($L$ = 3.6 μm, without $C_{mem}$ and $C_f$, $V_{DD}$ = 0.6 V), showing a response time of less than 1 ms: input signal on (**c**) and input signal off (**d**).

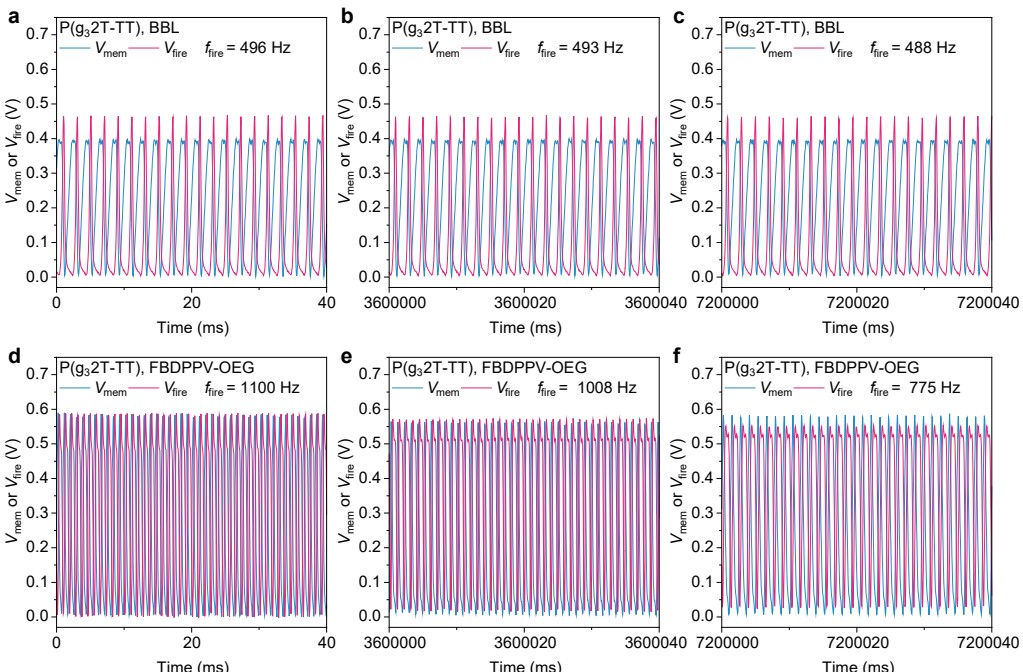

**Extended Data Fig. 8 | Long-term firing stability. a-c**, Continuous spiking behavior of OECNs based on P(g$_3$2T-TT)/BBL ($L$ = 3.6 μm, without $C_{mem}$ and $C_f$, $V_{DD}$ = 0.6 V) over two hours: 0 h (**a**), 1 h (**b**) and 2 h (**c**). **d-f**, Continuous spiking behavior of OECNs based on P(g$_3$2T-TT)/FBDPPV-OEG ($L$ = 3.6 μm, without $C_{mem}$ and $C_f$, $V_{DD}$ = 0.6 V) over two hours: 0 h (**d**), 1 h (**e**) and 2 h (**f**).

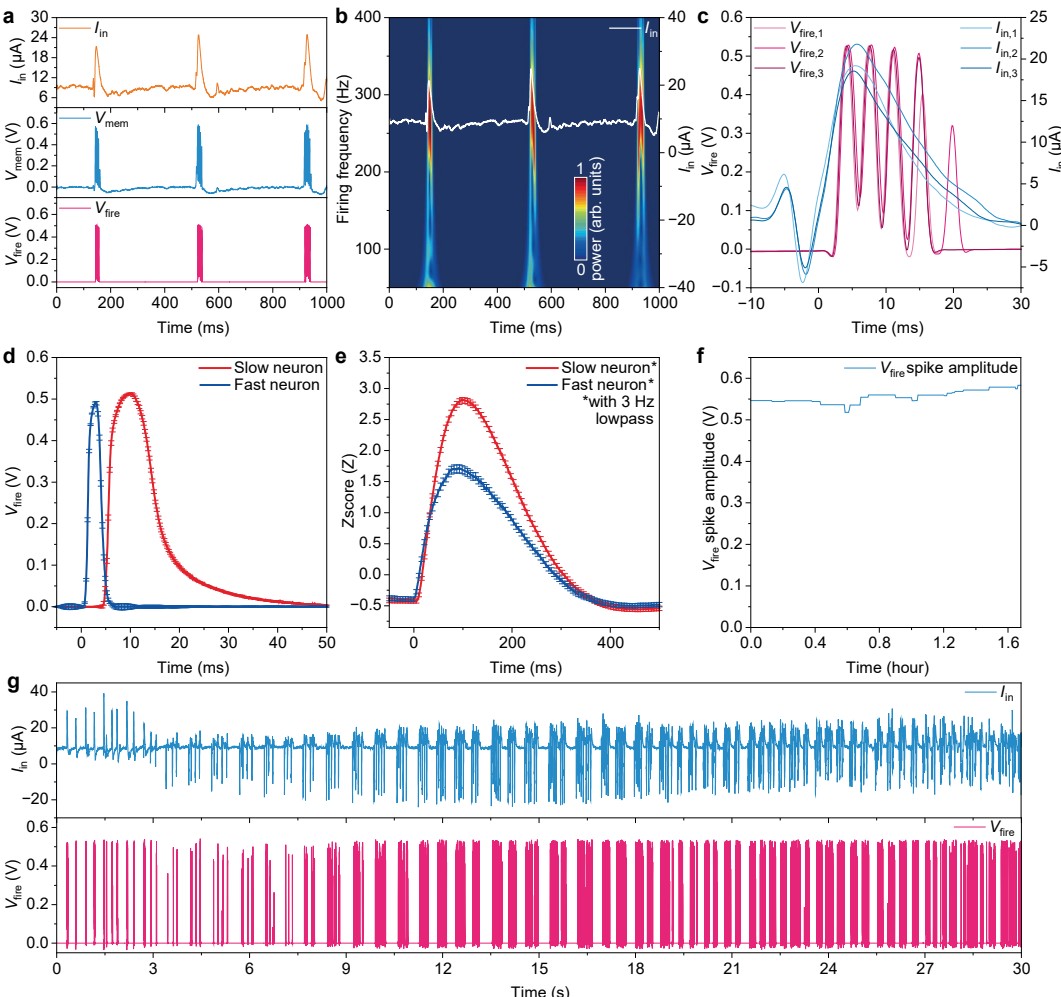

**Extended Data Fig. 9 | Closed-loop control with OECNs. a**, Sample trace of a hippocampal epileptiform discharge and corresponding evoked OECN firing. **b**, Spectra of $V_{out}$ during epileptiform-discharge-evoked neuron firing. **c**, Sample traces of OECN's voltage response (pink trace) towards epileptiform discharges (light blue trace). **d-e**, OECN responses to IEDs: (**d**), $V_{fire}$ in response to IEDs of a

400 Hz OECN (blue) and a 71 Hz OECN (red); and (**e**) 200 ms Gaussian waveform generated by 3 Hz low-pass filtering of $V_{fire}$ (**e**). **f**, Long recording period of OECN spiking amplitude. **g**, OECN firing during seizure onset. Error bars represent the standard deviation. n = 5 independent experiments.

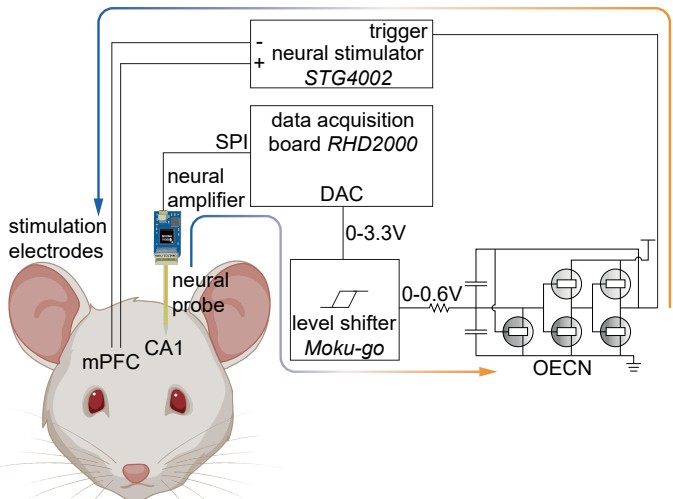

**Extended Data Fig. 10 | Closed-loop control with OECNs.** Schematic of the signal flow from the brain through neural amplifier, OECN, and stimulator.

Dion Khodagholy

# Reporting Summary

## Statistics

For all statistical analyses, confirm that the following items are present in the figure legend, table legend, main text, or Methods section.

| n/a | Confirmed | |
|---|---|---|
| ☐ | ☒ | The exact sample size (*n*) for each experimental group/condition, given as a discrete number and unit of measurement |
| ☐ | ☒ | A statement on whether measurements were taken from distinct samples or whether the same sample was measured repeatedly |
| ☒ | ☐ | The statistical test(s) used AND whether they are one- or two-sided<br>*Only common tests should be described solely by name; describe more complex techniques in the Methods section.* |
| ☒ | ☐ | A description of all covariates tested |
| ☒ | ☐ | A description of any assumptions or corrections, such as tests of normality and adjustment for multiple comparisons |
| ☐ | ☒ | A full description of the statistical parameters including central tendency (e.g. means) or other basic estimates (e.g. regression coefficient) AND variation (e.g. standard deviation) or associated estimates of uncertainty (e.g. confidence intervals) |
| ☒ | ☐ | For null hypothesis testing, the test statistic (e.g. *F*, *t*, *r*) with confidence intervals, effect sizes, degrees of freedom and *P* value noted<br>*Give P values as exact values whenever suitable.* |
| ☒ | ☐ | For Bayesian analysis, information on the choice of priors and Markov chain Monte Carlo settings |
| ☒ | ☐ | For hierarchical and complex designs, identification of the appropriate level for tests and full reporting of outcomes |
| ☒ | ☐ | Estimates of effect sizes (e.g. Cohen's *d*, Pearson's *r*), indicating how they were calculated |

*Our web collection on statistics for biologists contains articles on many of the points above.*

## Software and code

Policy information about availability of computer code

| Data collection | Data were collected using instrument interfacing software provided by the manufacturers including Keysight quick IV, Liquid Instruments, Intan Technologies and National Instruments. |
|---|---|
| Data analysis | All data were analyzed using MATLAB 2021b. |

For manuscripts utilizing custom algorithms or software that are central to the research but not yet described in published literature, software must be made available to editors and reviewers. We strongly encourage code deposition in a community repository (e.g. GitHub). See the Nature Portfolio guidelines for submitting code & software for further information.

## Data

Policy information about availability of data

All manuscripts must include a data availability statement. This statement should provide the following information, where applicable:
- Accession codes, unique identifiers, or web links for publicly available datasets
- A description of any restrictions on data availability
- For clinical datasets or third party data, please ensure that the statement adheres to our policy

Data supporting the findings of this study are available in the paper and the Extended figures and Supplementary Information files. The data generated in this study are provided in the Source Data files. Source data are provided with this paper.

# Research involving human participants, their data, or biological material

Policy information about studies with human participants or human data. See also policy information about sex, gender (identity/presentation), and sexual orientation and race, ethnicity and racism.

| | |
|---|---|
| Reporting on sex and gender | NA |
| Reporting on race, ethnicity, or other socially relevant groupings | NA |
| Population characteristics | NA |
| Recruitment | NA |
| Ethics oversight | NA |

Note that full information on the approval of the study protocol must also be provided in the manuscript.

# Field-specific reporting

Please select the one below that is the best fit for your research. If you are not sure, read the appropriate sections before making your selection.

☒ Life sciences          ☐ Behavioural & social sciences          ☐ Ecological, evolutionary & environmental sciences

For a reference copy of the document with all sections, see nature.com/documents/nr-reporting-summary-flat.pdf

# Life sciences study design

All studies must disclose on these points even when the disclosure is negative.

| | |
|---|---|
| Sample size | All sample size have been reported in the figure legends |
| Data exclusions | No data is excluded |
| Replication | The number of replications per measurement, device, sample and trial are reported in figure legends |
| Randomization | NA |
| Blinding | NA |

# Reporting for specific materials, systems and methods

We require information from authors about some types of materials, experimental systems and methods used in many studies. Here, indicate whether each material, system or method listed is relevant to your study. If you are not sure if a list item applies to your research, read the appropriate section before selecting a response.

## Materials & experimental systems

| n/a | Involved in the study |
|---|---|
| ☒ ☐ | Antibodies |
| ☒ ☐ | Eukaryotic cell lines |
| ☒ ☐ | Palaeontology and archaeology |
| ☐ ☒ | Animals and other organisms |
| ☒ ☐ | Clinical data |
| ☒ ☐ | Dual use research of concern |
| ☒ ☐ | Plants |

## Methods

| n/a | Involved in the study |
|---|---|
| ☒ ☐ | ChIP-seq |
| ☒ ☐ | Flow cytometry |
| ☒ ☐ | MRI-based neuroimaging |

# Animals and other research organisms

Policy information about studies involving animals; ARRIVE guidelines recommended for reporting animal research, and Sex and Gender in Research

| | |
|---|---|
| Laboratory animals | Rats |
| Wild animals | NA |
| Reporting on sex | All the experimental rats are male. But sex is not a biological variable. We conducted neurophysiological recording to validate electronic devices. |
| Field-collected samples | NA |
| Ethics oversight | Columbia University IACUC, Assurance ID: D16-00003 |

Note that full information on the approval of the study protocol must also be provided in the manuscript.

# Plants

| | |
|---|---|
| Seed stocks | NA |
| Novel plant genotypes | NA |
| Authentication | NA |

