## [Peer Review File · Nature Sensors]

High-frequency, low-energy organic event-based sensors for closed-loop neurostimulation

Corresponding Author: Professor Simone Fabiano

Version 0:

Reviewer comments:

Reviewer #1

(Remarks to the Author)

This paper reports on a novel Organic Electrochemical Neuron (OECN) based on optimized materials and device architecture. Its core advantages lie in achieving an unprecedentedly high operating frequency (up to 1.1 kHz) and ultra-low energy consumption (as low as 38 pJ/spike), which is comparable to that of biological neurons. The authors first systematically demonstrate that by reducing the OECT channel length (to as small as 3.6 μm) and employing high-performance n-type materials (FBDPPV-OEG), it is possible to significantly enhance the device's switching speed and operating frequency while maintaining low-voltage (0.1 V) operation. Subsequently, the study demonstrates the application of this OECN in an animal model of epilepsy. The results indicate that the device outperforms conventional digital signal processing methods in accurately detecting Interictal Epileptiform Discharges (IEDs) and can drive a closed-loop system that suppresses IED-induced pathological activity via electrical stimulation.

I found this work overall interesting but the following questions should be addressed before publication.

1. In Figure 1d, it is suggested that connection points in the circuit diagram be indicated with solid black dots to improve readability. This is a standard practice in circuit schematics and would help avoid ambiguity regarding node connections, especially in more complex configurations.
2. In Figures 2d and 2e, it would be helpful to explicitly label the switching voltages or threshold levels. Doing so would aid readers in understanding the voltage transitions involved in the circuit's operation, particularly in relation to the firing and reset phases described in the text.
3. The proposed OECN operates as a current-mode sensor, integrating input current on capacitor C1 until a sufficient charge is accumulated to trigger an output level inversion. The subsequent discharge path is controlled by the reset n-OECT, which pulls down the input node and shapes the output into a pulse waveform. This operational mechanism assumes that epileptic activity in the hippocampus generates sufficiently strong current signals to drive the input node. However, in the manuscript, IEDs (interictal epileptic discharges) are defined as voltage signals (Page 11, Line 2), and there is no clarification regarding their actual nature—whether they are primarily voltage or current signals in the biological context. This inconsistency may confuse readers. It is recommended to include a brief discussion on the electrophysiological nature of IEDs, supported by relevant references, to justify the use of current as the input signal modality for the sensor.
4. In Figure 1d, the definitions of capacitors C1 and C2 remain somewhat ambiguous. In particular, the term "membrane capacitor" is used without a clear explanation of its physical meaning—what it refers to in the device structure, how it is formed, and whether it is an inherent or tunable component. Since both C1 and C2 directly determine the time constant of the current integration process, their precise role and origin are critical to understanding the circuit's behavior. It is therefore recommended to include a detailed interface circuit model at the I_{in} and V_{fire} ports. Furthermore, the manuscript should explicitly explain the physical mechanisms behind the formation of C1 and C2 (e.g., electrostatic double-layer capacitance, gate oxide capacitance, or parasitic contributions), as well as practical methods for adjusting or tuning their values.
5. Considering the limited volume of brain regions like the hippocampus, what was the specific size of the implant used in the in vivo experiments? Could the device implantation elicit an inflammatory response in the surrounding tissue? While the authors emphasize the material's flexibility and organic nature when discussing biocompatibility, there is no in-depth discussion of the physical impact of the implant's footprint or its long-term stability (e.g., fibrosis, corrosion).
6. In freely moving animal models, movement inevitably causes electrode displacement, introducing motion artifacts into the neural recordings. These artifacts, whose frequency and amplitude can sometimes resemble pathological signals like IEDs,

are a common source of interference in closed-loop detection systems. How does the OECN detection system perform while the animal is awake and moving freely? How does the system differentiate between genuine IED signals and motion artifacts? Could strong motion artifacts lead to false positives, triggering unnecessary electrical stimulation? This would not only compromise the therapeutic efficacy but could also induce side effects.

7. The flexible OECN array shown in Figure 3i is compact, but does it represent the complete implantable device? Beyond this array, what other back-end components (e.g., power management, wireless communication) comprise the full system? The current setup appears to be a "wired" proof-of-concept system, as the methods describe the use of external power supplies and signal processors.

8. Epileptic seizures are diverse in their manifestation, and IEDs are only biomarkers of the interictal period. The neural activity patterns during the ictal period (e.g., high-frequency, high-amplitude bursting discharges) differ significantly from IEDs. While the paper notes that OECNs can detect ictal activity, what is the therapeutic efficacy of this closed-loop system and its stimulation strategy in other epilepsy models, such as an acute seizure model?

Reviewer #2

(Remarks to the Author)

Manuscript Number: NATSENSORS-25050320

The manuscript presents an organic electrochemical neuron (OECN)-based sensor designed for real-time, energy-efficient detection of neural signals, with potential application in closed-loop neurostimulation. The proposed sensors are event-driven, responding within approximately 1 ms and generating voltage pulses up to 1.1 kHz—adequately covering the bandwidth of mammalian neuronal activity (0.5–1000 Hz). Impressively, they operate at an energy cost as low as ~40 pJ per spike. This enables the detection of hippocampal interictal epileptiform discharges (IEDs) with performance reportedly exceeding that of conventional detection methods. Furthermore, the integration of these sensors with microelectrodes allows for real-time stimulation to suppress pathological sleep spindle oscillations in vivo. While the topic is timely and the proposed technology holds promise for future implantable bioelectronic systems, the manuscript lacks the depth, critical analysis, and scientific rigor expected for publication in Nature Sensors. The current version does not sufficiently contextualize the work within the existing literature, nor does it provide the level of technical insight and validation necessary to support its claims. Therefore, I do not recommend this manuscript for publication in Nature Sensors:

#1: The introduction is overly brief and does not sufficiently establish the main problem addressed by the work. The study focuses on the development of high-frequency, low-energy OECN-based sensors, where the optimization of device architecture and materials is a critical aspect. However, the introduction lacks a thorough review of the materials employed in such systems as well as the optimization strategies for device architecture. A more comprehensive discussion is necessary to clearly define the research focus, emphasize its significance, and provide adequate background.

#2: The font size of the text in all the figures is too small to read.

#3: After an abbreviation has been introduced, the full form should not be repeated.

#4: In this work, glycolated polythiophene p(g 3 2T-TT) is used as the channel material for p-type OECTs, while poly(benzimidazobenzophenanthroline) (BBL) and glycolated fluorinated benzodifurandione-based poly(p-phenylene vinylene) (FBDPPV-OEG) are employed for n-type OECTs. The authors state that these materials were selected due to their high stability and favorable OECT performance. However, this brief justification is insufficient. A more detailed rationale, including a comparative table summarizing key performance parameters of

similar materials, is necessary to support the material choices and highlight their advantages.

#5: The thickness of the channel layer plays a critical role in determining OECT device performance; however, the manuscript does not address the optimization of this parameter.

#6: An energy level diagram comparing the energy levels of the different device layers is missing from the manuscript.

#7: Supplementary Fig. 23: The author claims that the figure presents a comparison of OECN power consumption with other closed-loop neural signal processors. However, the graph lacks comprehensiveness, and the reference numbers corresponding to each data point are missing.

#8: In the introduction, the author mentions the optimization of materials; however, in OECTs, the electrolyte plays a crucial role. The manuscript does not address the optimization of electrolyte concentration.

#9: The conclusion should be more informative.

Reviewer #3

(Remarks to the Author)

Yang et al. report on the development of a high-frequency, low-energy consumption organic electrochemical neuron

(OECN)-based sensors for closed-loop neurostimulation. The synergistic optimization of device architecture and materials yields event-driven sensors with ~1 ms response, 1.1 kHz bandwidth, and ~40 pJ energy per spike, matching mammalian neuron dynamics. On this basis, they integrate ultrafast, low-power OECNs into a closed-loop neurostimulation system that detected IEDs and delivered real-time stimulation to suppress epileptic activity. This demonstrates their potential as scalable, biocompatible, and energy-efficient alternatives for next-generation neural interfaces and neuromorphic devices. The paper is well organized thoroughly. The manuscript needs to be revised to be ready for publication.

1. On Page 6, the authors claim that "These OECTs are at least 4 times faster than previously reported planar OECTs." However, the transient response is still slower than vertical or internal-gated OECTs. Please explain the advantages of using planar devices. Can we further improve the OECN device performance through the optimization of device structure?
2. On Page 6, the authors state that "a VDD level above 0.5 V is preferred to minimize hysteresis." Since voltage can influence the device's response speed, we are curious whether the n-type and p-type materials exhibit limitations or mismatches at different voltages, which could impact the overall response time. We recommend including an analysis of the response times of the two types of materials across a range of voltages.
3. On Page 8, the values of C1 and C2 are reported as 0 nF. Could the authors clarify why these capacitances are zero in certain areas? Additionally, is this phenomenon due to compensation effects from other components in the system?
4. The relationship between response time and input time constant is summarized in Figure 4d. We would like to know how to input a time constant of 10^{-4} ms and what is the fastest response time?
5. As demonstrated in this work, the p(g32T-TT)/FBDPV-OEG system is superior to p(g32T-TT)/BBL, with lower energy consumption. Could the authors elaborate on the underlying reasons for this difference? Additionally, could they provide suggestions for the selection of p- and n-channel materials to further optimize device performance?
6. Meanwhile, this study advances the development of OECN devices that meet the requirements of mammalian neuronal activity through optimization of device structure and materials. However, the unique advantages of the proposed design and optimization strategy are not clearly articulated. Could the authors further emphasize the distinctive features of this work?

Version 1:

Reviewer comments:

Reviewer #1

(Remarks to the Author)

The revised manuscript provides a well-documented characterization of the electrical performance of the OECN devices, clearly demonstrating their advantages.

However, the current work on in vivo closed-loop neuromodulation appears to be somewhat limited. The current study presents a proof-of-concept demonstration, showing the feasibility of using OECN as an external computing unit, rather than being integrated into an implantable closed-loop neuromodulation system.

As the authors note in the abstract, "While silicon-based neural interfaces offer high precision and speed, their rigidity and high power demand hinder long-term bio-integration," highlighting the importance of flexibility and low power for bio-integration. While the OECN device indeed offers flexibility and low power, it is not integrated in vivo but only functions in vitro, limiting the relevance of these advantages. The benefits of flexibility and low power are primarily intended to minimize biological tissue damage, yet this justification becomes less meaningful when the device is used outside the body, especially when compared to commercially available chips.

In the Methods section, the authors state that rigid tungsten wires are used as implanted electrodes. Neural signals are recorded by external equipment, then passed to the OECN for processing. The OECN output is converted and transmitted to a commercial stimulator, which finally delivers stimulation via implanted electrodes. In this setup, the OECN functions as an intermediate processing module, and its operational environment, such as its electrolyte, its volume, and feasibility of in vivo integration, remains unclear. The connectivity among system components is also insufficiently displayed. It is recommended that the authors provide photographs of the closed-loop system to clarify these points.

Compared to fully implantable closed-loop systems like DBS (Deep Brain Stimulation) or RNS (Responsive Neurostimulation) devices, the specific advantages of the OECN-based approach remain to be clarified.

Reviewer #2

(Remarks to the Author)

Manuscript Number: NATSENSORS-25050320A

The authors have addressed all reviewer comments in the response letter, and made considerable revisions to the manuscript. All reviewers and myself have raised some key concerns which were effectively and convincingly addressed, including clarification of circuit elements and device dimensions and biocompatibility. In vivo experimental setup, artifact mitigation, and the electrophysiological rationale of the IED signals were also resolved. Ambiguous terms were refined, relevant new content added with proper citations, and the figures were improved for clarity. The revised manuscript contains a broader scope, openly acknowledges current limitations such as the wired nature of the setup, and provides an integrated plan for chronic application. The reproducibility and technical quality of the work, including clarity, has improved significantly in published form. In this case I would suggest final acceptance, as all concerns have been thoroughly addressed and no major scientific issues remain.

Reviewer #3

(Remarks to the Author)

The authors have addressed the key issues with the first suggested revision, and the manuscript is now suitable for publication in Nature Sensors.

Version 2:

Reviewer comments:

Reviewer #1

(Remarks to the Author)

The authors have satisfactorily addressed my concerns, and I recommend accepting the manuscript for publication in Nature Sensors.

Response to the Reviewers

Dear referees, we found your reviews to be very thoughtful, and the comments were extremely helpful in enhancing the quality and thus the impact of our manuscript. Below, please find our point-by-point response in red lettering to your concerns and a description of how and where revisions to the manuscript have been made.

Referee #1 (Remarks to the Author):

This paper reports on a novel Organic Electrochemical Neuron (OECN) based on optimized materials and device architecture. Its core advantages lie in achieving an unprecedentedly high operating frequency (up to 1.1 kHz) and ultra-low energy consumption (as low as 38 pJ/spike), which is comparable to that of biological neurons. The authors first systematically demonstrate that by reducing the OECT channel length (to as small as 3.6 μm) and employing high-performance n-type materials (FBDPPV-OEG), it is possible to significantly enhance the device's switching speed and operating frequency while maintaining low-voltage (0.1 V) operation. Subsequently, the study demonstrates the application of this OECN in an animal model of epilepsy. The results indicate that the device outperforms conventional digital signal processing methods in accurately detecting Interictal Epileptiform Discharges (IEDs) and can drive a closed-loop system that suppresses IED-induced pathological activity via electrical stimulation.

I found this work overall interesting but the following questions should be addressed before publication.

We thank the reviewer for their very positive commentary on our manuscript. In the following, we addressed his/her remarks.

1. In Figure 1d, it is suggested that connection points in the circuit diagram be indicated with solid black dots to improve readability. This is a standard practice in circuit schematics and would help avoid ambiguity regarding node connections, especially in more complex configurations.

Excellent suggestion. We have added connection points in the circuit diagram of revised Fig. 1d, as shown below (see Fig. R1).

Figure R1. Updated circuit diagram of the LIF-type OECN.

2. In Figures 2d and 2e, it would be helpful to explicitly label the switching voltages or threshold levels. Doing so would aid readers in understanding the voltage transitions involved

in the circuit's operation, particularly in relation to the firing and reset phases described in the text.

Switching voltages and threshold levels have been clearly labeled in the revised Fig. 2d-e (see Fig. R2 below).

Figure R2. Updated schematic diagram of the working principle of the LIF-type OECN.

3. The proposed OECN operates as a current-mode sensor, integrating input current on capacitor $C1$ until a sufficient charge is accumulated to trigger an output level inversion. The subsequent discharge path is controlled by the reset n-OECT, which pulls down the input node and shapes the output into a pulse waveform. This operational mechanism assumes that epileptic activity in the hippocampus generates sufficiently strong current signals to drive the input node. However, in the manuscript, IEDs (interictal epileptic discharges) are defined as voltage signals (Page 11, Line 2), and there is no clarification regarding their actual nature—whether they are primarily voltage or current signals in the biological context. This inconsistency may confuse readers. It is recommended to include a brief discussion on the electrophysiological nature of IEDs, supported by relevant references, to justify the use of current as the input signal modality for the sensor.

Excellent observation, we thank the reviewer for pointing this out. We agree that the original manuscript did not adequately discuss the nature of IEDs, and we have now clarified this in the revised text. Electrophysiological signals, including IEDs, are typically measured as a potential difference (voltage) between a recording and a reference electrode. IEDs, like other local field potentials, arise from the summation of synaptic activity in large populations of neurons, with amplitudes in the range 0.1-1 mV, often larger than those of healthy physiological signals. Due to the relatively high impedance of the electrode-tissue interface and the low amplitude of these signals, amplification is required before processing. In our setup, the IED signals were first amplified using a preamplifier and then converted into an input current for the OECNs via a 100 k Ω series resistor. The OECN then integrates this input current and processes it in a way analogous to a biological neuron. We have added a brief discussion clarifying this point on p. 11 of the revised manuscript.

4. In Figure 1d, the definitions of capacitors C_1 and C_2 remain somewhat ambiguous. In particular, the term "membrane capacitor" is used without a clear explanation of its physical meaning—what it refers to in the device structure, how it is formed, and whether it is an inherent or tunable component. Since both C_1 and C_2 directly determine the time constant of the current integration process, their precise role and origin are critical to understanding the circuit's behavior. It is therefore recommended to include a detailed interface circuit model at the I_{in} and V_{fire} ports. Furthermore, the manuscript should explicitly explain the physical mechanisms behind the formation of C_1 and C_2 (e.g., electrostatic double-layer capacitance, gate oxide capacitance, or parasitic contributions), as well as practical methods for adjusting or tuning their values.

This is another excellent comment, which gives us the opportunity to clarify our statement. C_1 (or C_{mem}) is defined as the membrane capacitor, which mimics the membrane capacitance of biological neurons by integrating the input current (I_{in}) into the membrane voltage (V_{mem}). C_2 (or C_f) is defined as the feedback capacitor, which provides positive feedback, accelerating the rise of V_{fire} once the threshold is reached and shaping the spiking behavior of the circuit. Both C_1 and C_2 contribute to the total capacitance and thus directly determine the time constant of the integration and firing behavior. To better clarify their roles in the revised manuscript, we have replaced C_1 and C_2 with the more descriptive terms C_{mem} and C_f .

We have also performed SPICE simulations to illustrate how C_{mem} and C_f influence the spiking dynamics of LIF-type OECNs (see Fig. R3), summarized below:

Phase 1 – Integration: At the beginning of the spiking cycle, C_{mem} and C_f operate in parallel to integrate the input charge.

Phase 2 – Firing: When the threshold is reached, V_{fire} switches HIGH, and V_{mem} rises sharply as C_{mem} is charged by both I_{in} and feedback voltage through C_f .

Phase 3 – Reset: The HIGH V_{fire} turns on the reset OECT, which discharges C_{mem} .

Phase 4 – Recovery: When V_{mem} drops below the threshold, V_{fire} switches LOW, and V_{mem} rapidly discharges through both C_{mem} and C_f .

Removing C_f simplifies the charging and discharging dynamics to a single capacitor (C_{mem}), resulting in a higher firing rate due to the reduced total capacitance. The ratio of C_{mem} to C_f can also be tuned to adjust the dynamics: *i*) Larger C_f results in steeper rise and fall in V_{mem} (faster phases 2 & 4); *ii*) Larger C_{mem} , results in longer integration and reset phases (phases 1 & 3).

In actual devices, C_{mem} and C_f can be implemented using an electrolyte between gold electrode gaps, yielding double-layer capacitances of ~ 1 -10 nF. Larger capacitance can be implemented using commercial capacitors. Alternatively, C_{mem} can simply rely on the parasitic capacitance of the OECTs (< 1 nF). We refer to this minimal configuration without integrated C_{mem} and C_f as 'without C_{mem} and C_f ' or 'w/o C_{mem} and C_f ' in the Figure legends.

We have added the simulations and the discussion above to the revised Supplementary Fig. 1.

Figure R3. a, Simulated rising and falling dynamics of V_{mem} based on the SPICE model. b-c, Comparison of LIF neuron spiking behavior with (b) and without (c) C_f . d-e, Comparison of LIF neuron spiking with different C_{mem}/C_f ratios: 1:5 (d) and 5:1 (e).

5. Considering the limited volume of brain regions like the hippocampus, what was the specific size of the implant used in the in vivo experiments? Could the device implantation elicit an inflammatory response in the surrounding tissue? While the authors emphasize the material's flexibility and organic nature when discussing biocompatibility, there is no in-depth discussion of the physical impact of the implant's footprint or its long-term stability (e.g., fibrosis, corrosion).

We appreciate this insightful comment. For clarification, in this study the OECN circuit was not directly implanted into the brain but rather integrated with already established implantable technologies¹. This approach decouples the complexities of stereotaxic electrode implantation from the OECN, minimizes changes to the surgical procedure, and demonstrates the OECN's compatibility with already established procedures, thereby simplifying its translational path. As the reviewer rightly noted, the in vivo performance and long-term safety of neural implants and their electronics are critical for clinical translation. To address this, we used Parylene-C as a hermetic coating material. Parylene-C is an FDA-approved polymer that meets the USP Class

VI and ISO 10993 biocompatibility standards², and several studies have demonstrated its long-term safety and stability in implantable devices. We have revised the manuscript text (p. 13) to clarify these points and included additional discussion to highlight the clinical relevance and applicability of the OECN.

6. In freely moving animal models, movement inevitably causes electrode displacement, introducing motion artifacts into the neural recordings. These artifacts, whose frequency and amplitude can sometimes resemble pathological signals like IEDs, are a common source of interference in closed-loop detection systems. How does the OECN detection system perform while the animal is awake and moving freely? How does the system differentiate between genuine IED signals and motion artifacts? Could strong motion artifacts lead to false positives, triggering unnecessary electrical stimulation? This would not only compromise the therapeutic efficacy but could also induce side effects.

Excellent point regarding the practical application of OECN in closed-loop therapeutics. We want to emphasize that our earlier work identified non-rapid eye movement (NREM) sleep as the most effective epoch for intervention, owing to the neural synchrony and higher excitability of cortical circuits during this state^{1,3,4}. This choice also makes the approach less susceptible to motion artifacts. The reviewer is correct that the large amplitude and waveform profile of EMG signals and motion artifacts pose challenges for the accurate identification of IEDs in noisy environments. To address this, several established strategies, fully compatible with the OECN, can be employed:

Noise detection: A secondary “noise channel” located away from the IED source ensures that IEDs are detected only in the hippocampal channel, while noise appears in both.

Median removal: The median signal across multiple channels is subtracted from the detection channel to suppress common noise sources such as EMG.

Upper thresholding: Events exceeding the expected IED amplitude, characteristic of EMG and movement artifacts, are excluded by applying an upper threshold.

These strategies can also be combined for more robust performance in complex scenarios. We have added these details to the Conclusions section (p. 13/14), highlighting the scalability of OECNs to other brain-state recordings.

7. The flexible OECN array shown in Figure 3i is compact, but does it represent the complete implantable device? Beyond this array, what other back-end components (e.g., power management, wireless communication) comprise the full system? The current setup appears to be a "wired" proof-of-concept system, as the methods describe the use of external power supplies and signal processors.

At this stage, this study presents a proof-of-concept demonstration of the OECN as a core computing unit. The current setup is wired, relying on external power supplies, electrodes, and amplifiers, as described in the Methods. This choice was intentional to focus on validating the fundamental operation, scalability, and ability of the OECN to operate at biologically realistic frequencies. We agree with the reviewer that a complete implantable system would also require additional back-end components, such as power management, wireless communication, and encapsulation suitable for chronic implantation. These elements are well-established in the field of implantable electronics and can, in principle, be integrated with the OECN in future work. We have clarified in the revised manuscript that this study is a wired, proof-of-concept

demonstration of the OECN's functionality and potential, and that further engineering will be needed to achieve a fully autonomous, implantable system.

8. Epileptic seizures are diverse in their manifestation, and IEDs are only biomarkers of the interictal period. The neural activity patterns during the ictal period (e.g., high-frequency, high-amplitude bursting discharges) differ significantly from IEDs. While the paper notes that OECNs can detect ictal activity, what is the therapeutic efficacy of this closed-loop system and its stimulation strategy in other epilepsy models, such as an acute seizure model?

We appreciate this insightful comment. In this work, we focused on demonstrating the ability of OECN to act as a physiological signal processor/detector, and for the first time, show the potential of spiking circuits for closed-loop interventions. We agree with the reviewer that epilepsy is a very complex neurological disorder, and it is unlikely that one approach will cure all its comorbidities. Here, we focused on using OECNs to suppress the pathological coupling between the hippocampus and neocortex, thereby improving memory. Epilepsy therapeutics focused on eliminating seizures have had limited efficacy in modifying disease course and addressing these comorbidities, which can profoundly impair quality of life⁵⁻⁷. Epileptic networks predominantly exist in the interictal state, which contains aberrant dynamics and epileptiform patterns that interfere with physiological processes^{8,9}. During NREM sleep, the consolidation of episodic memory requires a precise correlation of hippocampal and cortical oscillations, including hippocampal sharp wave-ripples, the cortical slow oscillation, cortical spindles, and cortical ripples¹⁰⁻¹³. IEDs, a key pathological output of the interictal state, disrupt these critical interactions by initiating strong, precise temporal coupling with sleep spindles, which surpasses physiological ripple-spindle correlation³. IED-spindle coupling occurs in rodent models and human patients with focal epilepsy, establishing this phenomenon as a potential interictal therapeutic target^{4,14,15}. We have added this discussion to the Results section (p. 10) of the revised manuscript.

Referee #2 (Remarks to the Author):

Manuscript Number: NATSENSORS-25050320

The manuscript presents an organic electrochemical neuron (OECN)-based sensor designed for real-time, energy-efficient detection of neural signals, with potential application in closed-loop neurostimulation. The proposed sensors are event-driven, responding within approximately 1 ms and generating voltage pulses up to 1.1 kHz—adequately covering the bandwidth of mammalian neuronal activity (0.5–1000 Hz). Impressively, they operate at an energy cost as low as ~40 pJ per spike. This enables the detection of hippocampal interictal epileptiform discharges (IEDs) with performance reportedly exceeding that of conventional detection methods. Furthermore, the integration of these sensors with microelectrodes allows for real-time stimulation to suppress pathological sleep spindle oscillations in vivo. While the topic is timely and the proposed technology holds promise for future implantable bioelectronic systems, the manuscript lacks the depth, critical analysis, and scientific rigor expected for publication in Nature Sensors. The current version does not sufficiently contextualize the work within the existing literature, nor does it provide the level of technical insight and validation necessary to support its claims. Therefore, I do not recommend this manuscript for publication in Nature Sensors:

We thank the reviewer for their insightful comments, which have helped us to substantially improve the quality of our manuscript. Below, we address each of their remarks in detail.

#1: The introduction is overly brief and does not sufficiently establish the main problem addressed by the work. The study focuses on the development of high-frequency, low-energy OECN-based sensors, where the optimization of device architecture and materials is a critical aspect. However, the introduction lacks a thorough review of the materials employed in such systems as well as the optimization strategies for device architecture. A more comprehensive discussion is necessary to clearly define the research focus, emphasize its significance, and provide adequate background.

We thank the reviewer for this comment. We respectfully disagree that the introduction was overly brief or failed to establish the main problem. In the original manuscript, we outlined the limitations of current silicon-based bioelectronics, described the advantages and challenges of OMIEC-based OECNs, and highlighted the need for high-frequency, low-energy, closed-loop devices for biomedical applications. Nevertheless, to further strengthen the introduction and to address the reviewer's suggestion, we have revised the text to explicitly acknowledge recent developments in materials and device architectures that improved OECN performance, while clarifying that existing devices still fall short of the biorealistic spiking frequencies, energy efficiency, and scalability required for real-time neuromodulation. We believe the revised introduction now more clearly situates our work in the context of prior art and emphasizes the specific challenges our work addresses, while remaining within the concise format expected for Nature-family journals.

#2: The font size of the text in all the figures is too small to read.

We have revised the figure font to comply with the formatting standards of Nature journals (<180 mm figure width, 5-7 pt font size).

#3: After an abbreviation has been introduced, the full form should not be repeated.

We thank the reviewer for the helpful suggestion. We have ensured that each abbreviation is defined only once in the abstract, main text, and figure captions, in accordance with the Nature journals' guidelines.

#4: In this work, glycolated polythiophene p(g32T-TT) is used as the channel material for p-type OECTs, while poly(benzimidazobenzophenanthroline) (BBL) and glycolated fluorinated benzodifurandione-based poly(p-phenylene vinylene) (FBDPPV-OEG) are employed for n-type OECTs. The authors state that these materials were selected due to their high stability and favorable OECT performance. However, this brief justification is insufficient. A more detailed rationale, including a comparative table summarizing key performance parameters of similar materials, is necessary to support the material choices and highlight their advantages.

Excellent suggestion. Selecting materials for high-frequency spiking operation requires careful consideration of charge transport performance, threshold voltage, and transient response. We have added Supplementary Tables 1 & 2 in the revised Supplementary Information to highlight the advantages of the three materials chosen for this study.

#5. The thickness of the channel layer plays a critical role in determining OECT device performance; however, the manuscript does not address the optimization of this parameter.

We agree that thickness optimization of the channel material is critical for achieving balanced charge transport. We systematically optimized the thickness prior to OECN fabrication (see Fig. R4) and have included the results as Supplementary Fig. 6 in the revised Supplementary Information.

Figure R4. Channel thickness-dependent transfer curves of OECTs based on (a) P(g32T-TT), (b) BBL, and (c) FBDPPV-OEG. At optimized channel thicknesses (5 nm for P(g32T-TT), 20 nm for BBL, and 7 nm for FBDPPV-OEG), the p-type and n-type OECTs exhibited perfectly balanced charge transport.

#6. An energy level diagram comparing the energy levels of the different device layers is missing from the manuscript.

The energy level diagram of the different materials (see Fig. R5) has been added in the revised Supplementary Information as Supplementary Fig. 3.

Figure R5. HOMO and LUMO energy levels of the OECT channel materials used in this study, along with the Fermi levels of the source/drain and gate electrodes. The HOMO level of the p-type material and the LUMO levels of the n-type materials are well aligned with the electrode Fermi levels, consistent with the small V_{th} observed in the OECTs.

#7. *Supplementary Fig. 23: The author claims that the figure presents a comparison of OECN power consumption with other closed-loop neural signal processors. However, the graph lacks comprehensiveness, and the reference numbers corresponding to each data point are missing.*

We thank the reviewer for this excellent suggestion. In the revised Supplementary Information, we have added Supplementary Table 3, which provides the detailed data from this figure along with the corresponding references.

#8. *In the introduction, the author mentions the optimization of materials; however, in OECTs, the electrolyte plays a crucial role. The manuscript does not address the optimization of electrolyte concentration.*

We thank the reviewer for raising this point. We note that 0.1 M NaCl is a standard electrolyte for OECTs, with an ionic strength comparable to that of physiological saline. This makes it well-suited for bioelectronic applications, ensuring compatibility with biological systems and facilitating biosensing and neural interface studies. Nonetheless, to fully address the reviewer's comment, we did examine the performance of our three channel materials under different NaCl concentrations (see Fig. R6). For P(g₃2T-TT), the performance is relatively insensitive to the electrolyte concentration: lowering it to 0.01 M has a minimal effect, while increasing it to 1 M slightly reduces the maximum current and raises the threshold voltage. In contrast, BBL and FBDPPV-OEG show more noticeable shifts in threshold voltage ($\sim\pm 0.1$ V) and corresponding changes in current when the concentration is varied by an order of magnitude.

Figure R6. a-c, Effect of electrolyte concentration on the performance of OECTs based on (a) P(g₃2T-TT), (b) BBL, and (c) FBDPPV-OEG.

#9. *The conclusion should be more informative.*

We thank the reviewer for this suggestion. We have revised the Conclusions section to make it more informative by summarizing the key findings of the study, highlighting the advantages of the proposed OECN architecture, and outlining potential directions for future research and applications.

Referee #3 (Remarks to the Author):

Yang et al. report on the development of a high-frequency, low-energy consumption organic electrochemical neuron (OECN)-based sensors for closed-loop neurostimulation. The synergistic optimization of device architecture and materials yields event-driven sensors with ~1 ms response, 1.1 kHz bandwidth, and ~40 pJ energy per spike, matching mammalian neuron dynamics. On this basis, they integrate ultrafast, low-power OECNs into a closed-loop neurostimulation system that detected IEDs and delivered real-time stimulation to suppress epileptic activity. This demonstrates their potential as scalable, biocompatible, and energy-efficient alternatives for next-generation neural interfaces and neuromorphic devices. The paper is well organized thoroughly. The manuscript needs to be revised to be ready for publication.

We thank the reviewer for their very positive commentary on our manuscript. In the following, we addressed his/her remarks.

1. On Page 6, the authors claim that “These OECTs are at least 4 times faster than previously reported planar OECTs.” However, the transient response is still slower than vertical or internal-gated OECTs. Please explain the advantages of using planar devices. Can we further improve the OECN device performance through the optimization of device structure?

This is an excellent comment. We agree that the transient response of the planar OECTs in this study is slower than that of previously reported *vertical* and *internal-gated* OECTs. However, the choice of device architecture is guided by the specific requirements of the application, in this case, complementary circuits, patterned electrolytes, and stable, cross-talk-free operation. Below, we outline the rationale for using planar devices and discuss possible directions for improving OECN performance through device structure optimization.

1. Vertically stacked (sandwich) channels:

Although this architecture yields shorter channel lengths and therefore higher speeds, it suffers from significant leakage currents due to the proximity of source/drain electrodes. Additionally, fabricating dedicated gate electrodes and patterned electrolytes for cross-talk-free operation becomes highly challenging in this configuration^{16,17}.

2. Internal ion-gated transistors (IGTs):

IGTs exhibit inherently faster dynamics because ions only need to travel a short distance within the channel material to modulate its conductivity. However, high-performance accumulation-mode n-type materials for IGTs are still lacking, which limits this approach to p-type devices¹⁸⁻²⁰. We recognize the promise of this architecture and are actively collaborating to develop n-type-based IGTs, which could enable complementary circuits based on this concept.

3. Vertical-contact planar devices:

These devices represent a realistic next step, offering short channel lengths (<1 μm), dedicated gate electrodes, and patterned electrolytes on a scalable and manufacturable platform^{21,22}. Such designs could improve OECN performance while remaining compatible with complementary circuits and ensuring stable operation.

We have added this discussion to the revised manuscript (p. 14) and cited relevant prior work to place our study in the context of these developments.

2. On Page 6, the authors state that “a VDD level above 0.5 V is preferred to minimize hysteresis.” Since voltage can influence the device’s response speed, we are curious whether the n-type and p-type materials exhibit limitations or mismatches at different voltages, which

could impact the overall response time. We recommend including an analysis of the response times of the two types of materials across a range of voltages.

Again, excellent comment. We performed transient response measurements of all the materials under investigation at different applied voltages. As shown in Fig. R7, as the voltage decreases, the switching-on transient response becomes approximately $2.7\times$ slower for P(g₃2T-TT) and less than $1.3\times$ slower for FBDPPV-OEG, while it remains unchanged for BBL. This results in longer delays in the complementary circuit at lower voltages. We have added these new results to the revised Supplementary Information as Supplementary Fig. 10.

Figure R7. a-c, Switching-on transient characteristics of OECTs based on (a) P(g₃2T-TT), (b) BBL, and (c) FBDPPV-OEG at different V_D biases and V_G pulses. (d) Summary of the voltage-dependent transient response of the OECTs.

3. On Page 8, the values of C_1 and C_2 are reported as 0 nF. Could the authors clarify why these capacitances are zero in certain areas? Additionally, is this phenomenon due to compensation effects from other components in the system?

We thank the reviewer for pointing this out. We realize that the terminology we used may have caused confusion. What we meant with C_1 ($C_{\text{mem}} = C_2$ (C_f) = 0 nF is that no capacitors were used in the circuit. We have clarified this in the revised manuscript by rephrasing it as “w/o C_{mem} and C_f ”. In the absence of external C_{mem} and C_f , the parasitic capacitance of the OECTs (e.g., the gate-to-source capacitance of the resetting OECT and the gate capacitance of the first stage of the inverter) effectively acts as C_{mem} in the circuit.

4. The relationship between response time and input time constant is summarized in Figure 4d. We would like to know how to input a time constant of 10^{-4} ms and what is the fastest response time?

We thank the reviewer for bringing this error to our attention. The unit in the figure legend was incorrectly stated as milliseconds instead of seconds. This has now been corrected in the revised manuscript.

Figure R8. OECD response time as a function of input stage time constant.

5. As demonstrated in this work, the $p(g32T-TT)/FBDPV-OEG$ system is superior to $p(g32T-TT)/BBL$, with lower energy consumption. Could the authors elaborate on the underlying reasons for this difference? Additionally, could they provide suggestions for the selection of p - and n -channel materials to further optimize device performance?

Excellent comment. Due to its long polar side chains, FBDPPV-OEG exhibits more than $2\times$ lower volumetric capacitance (C^*) than BBL (see Fig. R9), while also offering over $3\times$ higher electron mobility. As a result, under comparable maximum source-drain current conditions, FBDPPV-OEG-based OECDs show significantly lower parasitic capacitance, leading to $2\times$ faster transient response and more than $2\times$ higher firing rate when combined with $P(g32T-TT)$ in OECDs.

In addition, the spiking duration is reduced by over 50%, while the power consumption of the second-stage inverter remains similar, resulting in significantly lower energy consumption per spike. FBDPPV-OEG also has a threshold voltage closer to that of $P(g32T-TT)$, enabling the OECDs to operate at a reduced supply voltage of 0.2 V, which further minimizes overall energy consumption. These results show that high charge carrier mobility, low threshold voltage, and moderate C^* are critical for fast OECD operation. Future research will focus on enhancing the stability of the active materials to extend the operational lifetime of the OECDs for long-term implantation.

We have added Fig. R9 as Supplementary Fig. S4 and incorporated the above discussion into the revised manuscript text (p. 6).

Figure R9. a-b, Volume-dependent capacitance of (a) BBL and (b) FBDPPV-OEG. The capacitance was measured using electrochemical impedance spectroscopy (EIS) at a voltage of -0.7 V . The volumetric capacitance was obtained from the linear fit of the volume-dependent data. FBDPPV-OEG exhibits more than two times lower C^* compared to BBL.

6. Meanwhile, this study advances the development of OECN devices that meet the requirements of mammalian neuronal activity through optimization of device structure and materials. However, the unique advantages of the proposed design and optimization strategy are not clearly articulated. Could the authors further emphasize the distinctive features of this work?

We thank the reviewer for this suggestion. We have revised the Conclusions section to make it more informative by clearly summarizing the key findings, emphasizing the unique advantages of the proposed OECN design and optimization strategy, and outlining promising directions for future research and applications.

Reference:

1. Ferrero, J. J. *et al.* Closed-loop electrical stimulation prevents focal epilepsy progression and long-term memory impairment. *Nat. Neurosci.* 1–10 (2025).
2. Golda-Cepa, M., Engvall, K., Hakkarainen, M. & Kotarba, A. Recent progress on parylene C polymer for biomedical applications: A review. *Prog. Org. Coatings* **140**, 105493 (2020).
3. Gelinás, J. N., Khodagholy, D., Thesen, T., Devinsky, O. & Buzsáki, G. Interictal epileptiform discharges induce hippocampal–cortical coupling in temporal lobe epilepsy. *Nat. Med.* 2016 226 **22**, 641–648 (2016).
4. Dahal, P. *et al.* Interictal epileptiform discharges shape large-scale intercortical communication. *Brain* **142**, 3502–3513 (2019).
5. Kwan, P. & Brodie, M. J. Early Identification of Refractory Epilepsy. *N. Engl. J. Med.* **342**, 314–319 (2000).
6. Perrine, K. *et al.* The Relationship of Neuropsychological Functioning to Quality of Life in Epilepsy. *Arch. Neurol.* **52**, 997–1003 (1995).
7. Boylan, L. S. *et al.* Depression but not seizure frequency predicts quality of life in treatment-resistant epilepsy. *Neurology* **62**, 258–261 (2004).
8. Kleen, J. K. *et al.* Hippocampal interictal epileptiform activity disrupts cognition in humans. *Neurology* **81**, 18–24 (2013).
9. Reed, C. M. *et al.* Extent of Single-Neuron Activity Modulation by Hippocampal Interictal Discharges Predicts Declarative Memory Disruption in Humans. *J. Neurosci.* **40**, 682–693 (2020).
10. Khodagholy, D., Gelinás, J. N. & Buzsáki, G. Learning-enhanced coupling between ripple oscillations in association cortices and hippocampus. *Science* **358**, 369–372 (2017).
11. Siapas, A. G. & Wilson, M. A. Coordinated Interactions between Hippocampal Ripples and Cortical Spindles during Slow-Wave Sleep. *Neuron* **21**, 1123–1128 (1998).
12. Girardeau, G., Benchenane, K., Wiener, S. I., Buzsáki, G. & Zugaro, M. B. Selective suppression of hippocampal ripples impairs spatial memory. *Nat. Neurosci.* **12**, 1222–1223 (2009).
13. Maingret, N., Girardeau, G., Todorova, R., Goutierre, M. & Zugaro, M. Hippocampo-cortical coupling mediates memory consolidation during sleep. *Nat. Neurosci.* 2016 197 **19**, 959–964 (2016).
14. Yu, H. *et al.* Interaction of interictal epileptiform activity with sleep spindles is associated with cognitive deficits and adverse surgical outcome in pediatric focal epilepsy. *Epilepsia* **65**, 190–203 (2024).
15. Sákovics, A. *et al.* Prolongation of cortical sleep spindles during hippocampal interictal epileptiform discharges in epilepsy patients. *Epilepsia* **63**, 2256–2268 (2022).
16. Kim, J. *et al.* Monolithically integrated high-density vertical organic electrochemical transistor arrays and complementary circuits. *Nat. Electron.* **7**, 234–243 (2024).
17. Huang, W. *et al.* Vertical organic electrochemical transistors for complementary circuits. *Nature* **613**, 496–502 (2023).
18. Spyropoulos, G. D., Gelinás, J. N. & Khodagholy, D. *Internal Ion-Gated Organic Electrochemical Transistor: A Building Block for Integrated Bioelectronics.* *Science Advances* vol. 5 (2020).
19. Cea, C. *et al.* Enhancement-mode ion-based transistor as a comprehensive interface and real-time processing unit for in vivo electrophysiology. *Nat. Mater.* **19**, 679–686 (2020).
20. Cea, C. *et al.* Integrated internal ion-gated organic electrochemical transistors for stand-alone conformable bioelectronics. *Nat. Mater.* 2023 2210 **22**, 1227–1235 (2023).
21. Yao, Y. *et al.* Flexible complementary circuits operating at sub-0.5 V via hybrid organic-inorganic electrolyte-gated transistors. *Proc. Natl. Acad. Sci. U. S. A.* **118**, e2111790118 (2021).

22. Wang, S. *et al.* A high-frequency artificial nerve based on homogeneously integrated organic electrochemical transistors. *Nat. Electron.* **8**, 254–266 (2025).

Response to the Reviewers

Dear referees, thank you for your thoughtful commentary on our manuscript. Below please find our point-by-point response in red lettering to your comments and a description of how/where revisions to the manuscript have been made.

Referee #1 (Remarks to the Author):

The revised manuscript provides a well-documented characterization of the electrical performance of the OECN devices, clearly demonstrating their advantages.

We thank the reviewer for acknowledging our efforts in delivering a comprehensive revision. We believe their constructive feedback has helped make the manuscript more accessible to a broader scientific audience. In the following, we addressed their remarks:

However, the current work on in vivo closed-loop neuromodulation appears to be somewhat limited. The current study presents a proof-of-concept demonstration, showing the feasibility of using OECN as an external computing unit, rather than being integrated into an implantable closed-loop neuromodulation system.

We agree with the reviewer that the present study represents a proof-of-concept demonstration rather than a fully implantable system. Nonetheless, our work constitutes the first realization of an organic spiking circuit that performs external computing for electrophysiological data processing, thereby demonstrating the feasibility of operating at the brain interface. Beyond achieving record-high electrical performance, this study highlights the potential of artificial spiking circuits for advancing neuromodulation strategies. We fully share the reviewer's view that there are many opportunities to expand this approach and ultimately translate it into an implantable closed-loop system that could benefit patients with neuropsychiatric disorders. We regard this as a highly promising direction for our ongoing research.

As the authors note in the abstract, "While silicon-based neural interfaces offer high precision and speed, their rigidity and high-power demand hinder long-term bio-integration," highlighting the importance of flexibility and low power for bio-integration. While the OECN device indeed offers flexibility and low power, it is not integrated in vivo but only functions in vitro, limiting the relevance of these advantages. The benefits of flexibility and low power are primarily intended to minimize biological tissue damage, yet this justification becomes less meaningful when the device is used outside the body, especially when compared to commercially available chips.

We would like to clarify the terminology: *in vivo* experiments are conducted in live organisms, while *in vitro* refers to experiments performed on extracted tissue placed in a dish. In our work, all experiments are performed on live, freely moving animals using chronic neural interface devices.

Currently, all active processing electronics are housed in an enclosure in responsive, closed-loop devices. This is necessary because they require batteries and power sources greater than 1 V, which increases the risk of hydrolysis and tissue damage. Our spiking circuit offers two distinct advantages:

Functional: Because they are made of low-voltage organic electrochemical transistors, they operate at lower voltages, and their spike-based computation consumes less overall current compared to traditional Si-based DSP systems.

Practical: Since they can be fabricated using conformable electronic processes rather than rigid packaging, they provide a more versatile and safer implant.

We would also like to emphasize that aqueous, ion-conducting environments induce crosstalk between transistors due to circulating fluids. Therefore, all processing transistors require some degree of insulation. In the case of Si-based transistors, this insulation is critical for device function because ion implantation can damage them. In contrast, electrochemical transistors are designed to function in aqueous environments. We have extensively explored various approaches in our previous publication (Cea et al., *Nat. Mater.*, 2023), including hydration vias and internal ion-gated strategies. Building on these advances, we are actively working toward integrating artificial spiking circuits directly into neural interfaces in future research.

In the Methods section, the authors state that rigid tungsten wires are used as implanted electrodes. Neural signals are recorded by external equipment, then passed to the OECN for processing. The OECN output is converted and transmitted to a commercial stimulator, which finally delivers stimulation via implanted electrodes. In this setup, the OECN functions as an intermediate processing module, and its operational environment, such as its electrolyte, its volume, and feasibility of in vivo integration, remains unclear. The connectivity among system components is also insufficiently displayed. It is recommended that the authors provide photographs of the closed-loop system to clarify these points. Compared to fully implantable closed-loop systems like DBS (Deep Brain Stimulation) or RNS devices, the specific advantages of the OECN-based approach remain to be clarified.

We have added a detailed signal flow diagram as Suppl. Fig. 30 in the revised Supplementary Information. We would like to emphasize that the OECN functions as a neural signal processor specifically designed for real-time detection of biomarkers. The experimental setups presented here were chosen to demonstrate the OECN's capability both as a processor and as a novel computational approach that can be applied to established surgical interventions. While the prospect of a fully implantable organic system, integrating a multi-electrode array, low-noise amplifiers, a multiplexer, a spike-based processor, and stimulators within a 4- μm PaC film, would indeed be highly attractive and revolutionary, it is beyond our current technological capabilities. In this work, our primary objective is to demonstrate the advantages of artificial spiking circuits for processing electrophysiological data.

Referee #2 (Remarks to the Author):

The authors have addressed all reviewer comments in the response letter, and made considerable revisions to the manuscript. All reviewers and myself have raised some key concerns which were effectively and convincingly addressed, including clarification of circuit elements and device dimensions and biocompatibility. In vivo experimental setup, artifact mitigation, and the electrophysiological rationale of the IED signals were also resolved. Ambiguous terms were refined, relevant new content added with proper citations, and the figures were improved for clarity. The revised manuscript contains a broader scope, openly acknowledges current limitations such as the wired nature of the setup, and provides an integrated plan for chronic application. The reproducibility and technical quality of the work, including clarity, has improved significantly in published form. In this case I would suggest final acceptance, as all concerns have been thoroughly addressed and no major scientific issues remain.

We thank the reviewer for the constructive suggestions provided in the previous round, which have helped us strengthen the manuscript.

Referee #3 (Remarks to the Author):

The authors have addressed the key issues with the first suggested revision, and the manuscript is now suitable for publication in Nature Sensors.

We thank the reviewer for the constructive suggestions provided in the previous round, which have helped us strengthen the manuscript.